# Number of Lines of Image Reconstructed from a Revealing Emission Signal as an Important Parameter of Rasterization and Coherent Summation Processes

Ireneusz Kubiak *, Artur Przybysz and Krystian Grzesiak

Department of Electromagnetic Compatibility, Military Communication Institute—State Research Institute, 05-130 Zegrze, Poland
* Correspondence: i.kubiak@wil.waw.pl

**Abstract:** An important issue in the protection of information against electromagnetic penetration is the possibility of its non-invasive acquisition. In many cases, getting hold of protected information involves recreating and presenting it in a readable and understandable form. In particular, this applies to data processed in graphic form and in such a form presented on the side of eavesdropping system. The effectiveness of reconstructing data in graphic form requires knowledge of raster parameters, i.e., the line length and the number of lines of the reproduced image. This article presents new measures allowing for the determination of the correct number of lines in an image. The maximum value of the measures has been proposed as a criterion for the correctness of determining the number of image lines. A predetermined number of image lines was assumed as the input data, which was determined on the basis of the analysis of the amplitude variability of the recorded revealing emission signal. The result of the considerations of the effectiveness of the measures adopted in the process of electromagnetic infiltration was the indication of methods that allow for the correct determination of the number of lines of the reproduced image. The correct number of image lines allows the use of the coherent summation algorithm of tens of images.

**Keywords:** security information; reveal emission; information processing; image processing; electromagnetic infiltration; data acquisition; contrast enhancement; image reconstruction

## 1. Introduction

Computerization of everyday life means that almost all information processed by us is in electronic form. The most popular devices in the information processing process are computers, laptops, laser printers, multifunctional devices, wireless communication terminals, etc. These devices have also become elements of extensive IT networks. Therefore, attention is paid to the protection of information in these types of networks very often, using the solutions to counteract cyberattacks [1–3].

Each of these devices, in accordance with the laws of physics, creates an electromagnetic field around it, which can change in time with changes in electrical signals in the form of which the information is processed. Thus, it becomes a source of electromagnetic emissions that spread uncontrollably around the device [4–10]. By using the physical properties of such a field, it is possible to come into possession of protected data without the knowledge of its owner (Figure 1). The most spectacular phenomenon is the emergence of emission sources during the processing of information in graphic form, e.g., a presentation of such information on various types of displays. Then, the acquired and reconstructed data can also be presented in the form of an image [11–15].

Due to unintended radiation, the recorded electromagnetic emissions are very often characterized by low levels. Hence, the quality of the reproduced and processed data in the form of images is very poor and the images require processing in order to extract essential information from them [16–20].

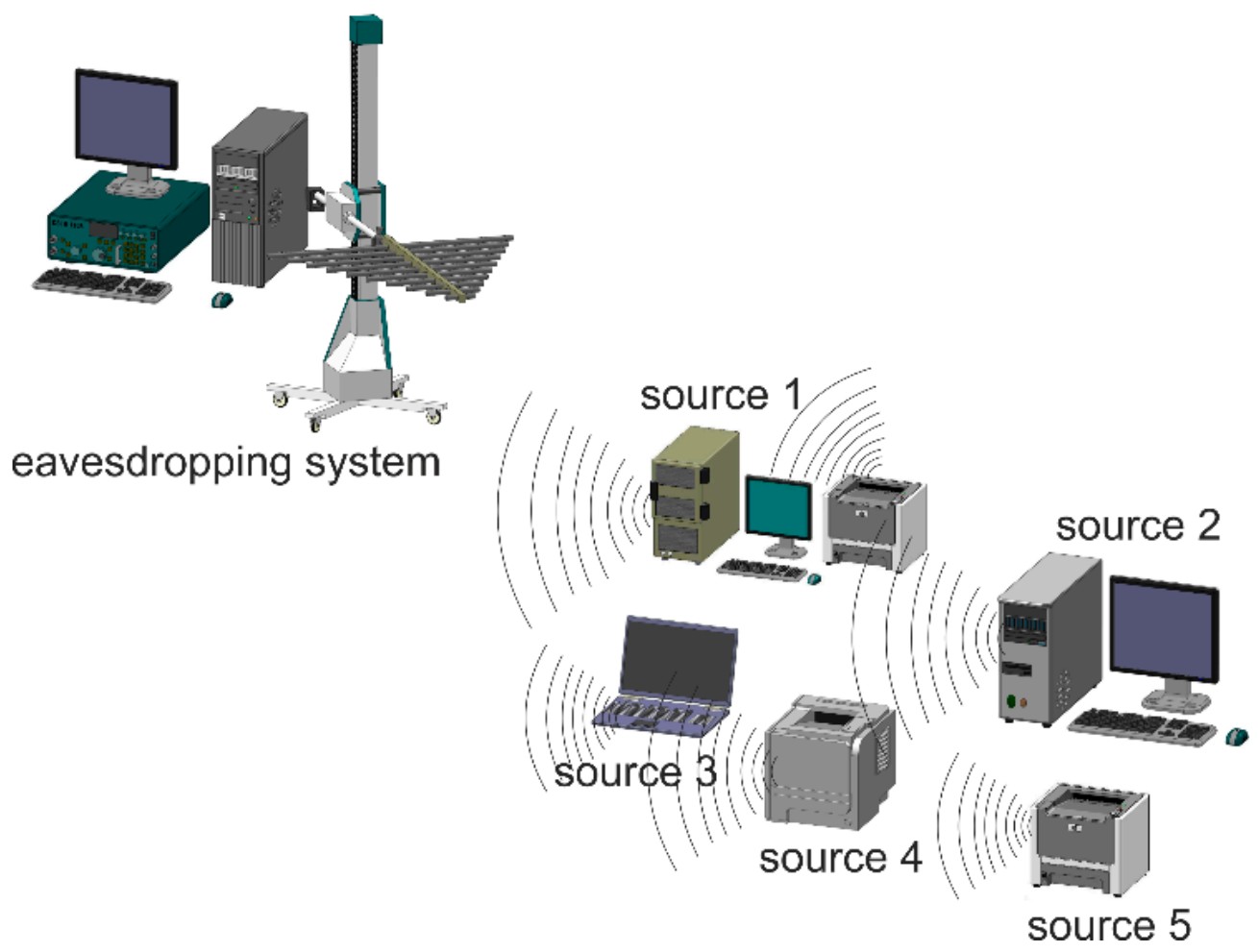

**Figure 1.** Surrounding us potential sources of undesirable emissions correlated with the information processed.

Devices intended for the processing of classified information require prior research in the scope of the assessment of measured electromagnetic emissions in laboratory conditions, i.e., in electromagnetically tight anechoic chambers(Figure 2). Meeting the requirements of information protection by the tested device inside the anechoic chamber guarantees information security in any other electromagnetic environment, in the environment surrounding us in particular.

The assessment is related to the determination of the degree of correlation of these emissions with the information processed [21–23]. This applies to devices that process data in graphic form in particular. The importance of this type of source of undesirable emissions results from the fact that the reconstructed data can be presented in the form of images containing human-readable and -understandable data [24–27]. In order for the reproduced data to be visualized in the form of images, it is necessary to know two basic image parameters: width (d image line length) and height (number $B_{Corr}$ of image lines) [28–31]. These parameters determine the correctness of the reconstruction of the image ready for effective further processing with the use of, e.g., the coherent summation algorithm of tens of realizations of the same image, which is possible thanks to a sufficiently long recorded implementation of the signal $s(t)$ of the revealing emission. It should be noted that in this case the emission source, which is the signal $x(t)$, must be a periodic signal in which the information about the processed data is repeated cyclically [30,31]. Such signals are, for example, video signals that excite graphic displays. Incorrect values of these parameters cause the quality of the reproduced image, after applying the image summation

process, to deteriorate and the data contained in it to become more unreadable (identifiable data becomes blurred, and not sharpened) or to result in an incomplete image (Figure 3).

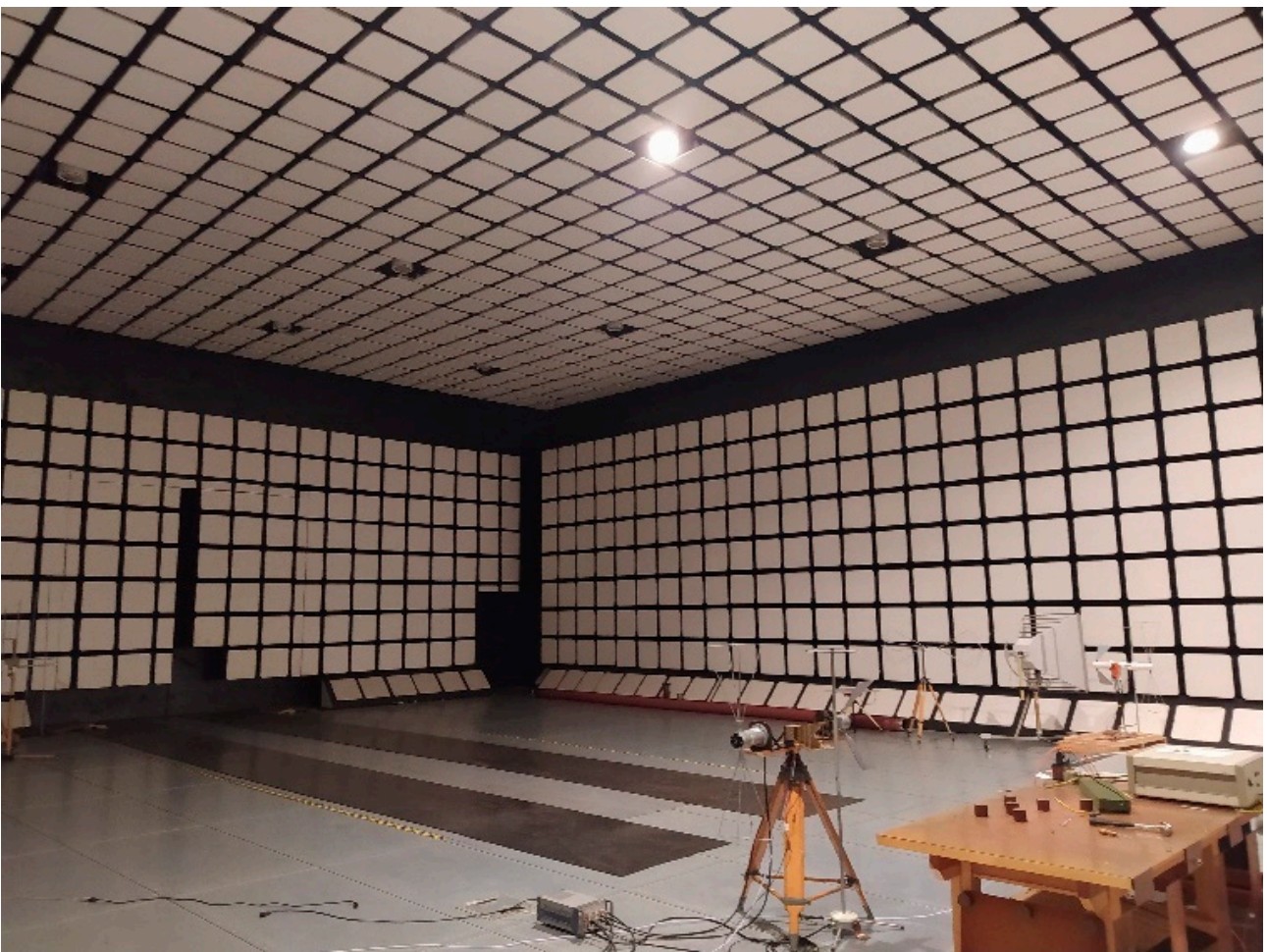

**Figure 2.** An example of anechoic chamber.

Determining the correct number $B_{Corr}$ of the reproduced image line results directly from its ignorance, which is related to a lack of access to the eavesdropped graphic imaging device in the electromagnetic infiltration process. This applies to typical computer monitors as well as any devices equipped with graphic displays, e.g., multifunctional devices. A rough estimate of the raster parameter of the number of lines in an image can be made based on a visual analysis of the reproduced image. The result of this analysis is an image that should not contain repeating graphic elements observed in a horizontal line. However, the exact indication of the number of image lines must already be carried out in an automatic manner, which allows for a quick classification of the recorded undesirable emissions.

At the same time, this process, together with the algorithm for determining the correct length d of the image line with an accuracy of $\Delta$ equal to at least $10^{-5}$, enables automation of the process of recreating graphic data in the form of images with the possibility of effective further summation of tens of realizations of the same image (Figure 4).

The method of pseudo-colouring of images was used to visualize the data. This made it possible to present the acquired data in the form of colour images, for which visual perception allows for the perception of more details than for images in grey colours [20,32]. Thus, the use of the pseudo-colouring algorithm facilitates the analysis and classification of revealing emissions.

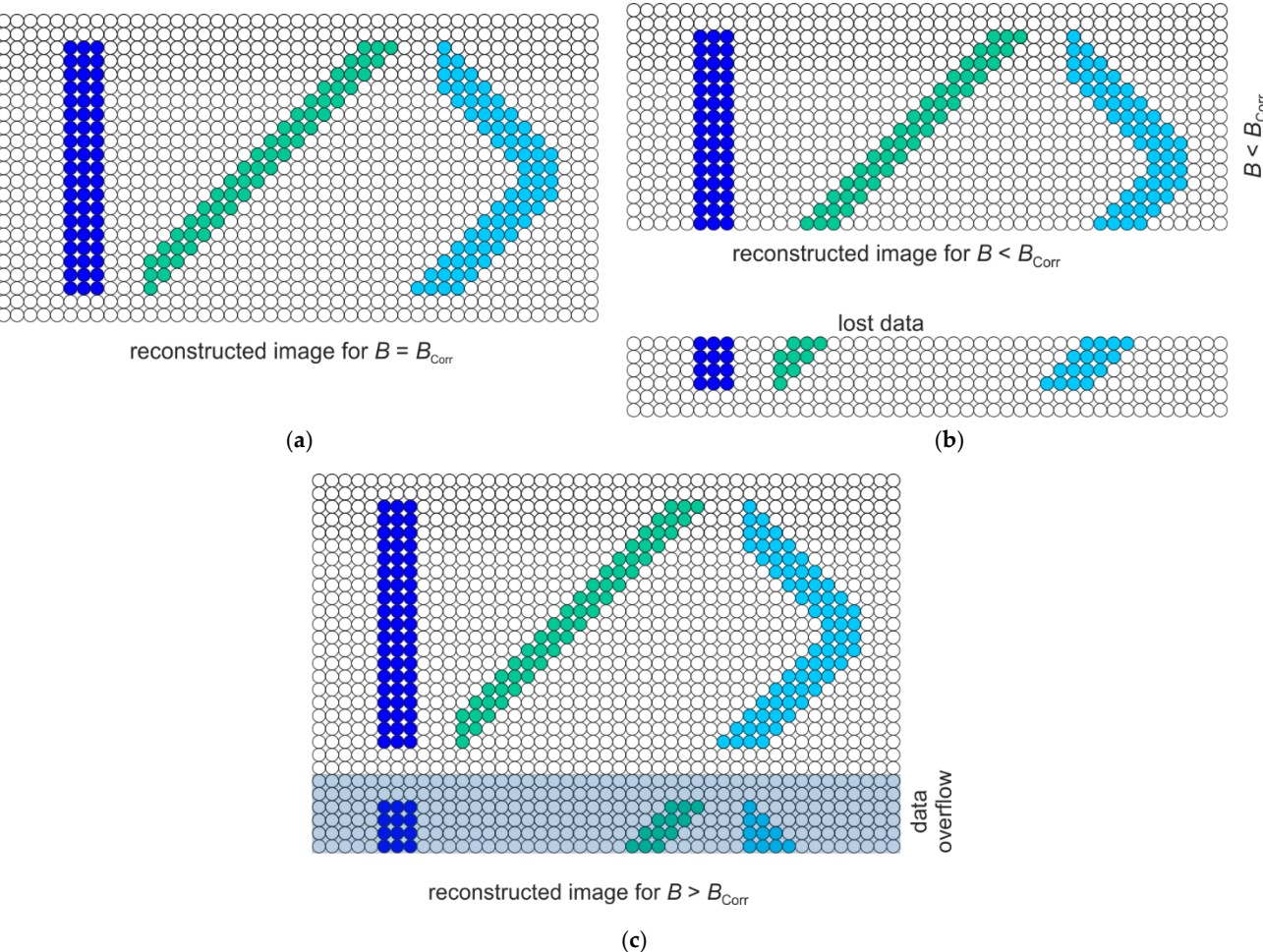

**Figure 3.** Illustrating the effect of the number lines of image on its content: (**a**) correct number lines of the image, (**b**) too few lines of the image, (**c**) too many lines of the image.

Incorrect determination of the number lines of image makes it impossible to effectively use image processing methods to improve its quality. In particular, it concerns the coherent summation algorithm of tens implementations of the same image (Figure 5).

One of the measures used in the assessment of image quality is the one related to the contrast of the analysed image [33]. We can mention here the measures based directly on the values of the maximum and minimum pixel amplitudes, the values of the average pixel amplitudes or the variance of the pixel amplitudes. The fulfilment of the appropriate criterion, i.e., the maximum value of these measures, clearly indicates the correct number $B$ of the line for which the appropriate values are calculated [28]. It should be noted, however, that the change in the number of B lines does not change the amplitudes of the pixels composing the image in the case of measures related to the contrast assessment. It only increases the number of pixels that are included in the calculation of each measure. Thus, it does not change the image quality and the contrast assessment. Hence, direct use of the measure of image contrast becomes ineffective. It also indicates the necessity to:

- Propose another dedicated measure and its criterion, effective in correctly determining the number of lines of the reconstructed image from the registered revealing broadcast signal;
- Pre-processing the reconstructed image depending on the B number of lines, and then analysing it in accordance with the adopted measure.

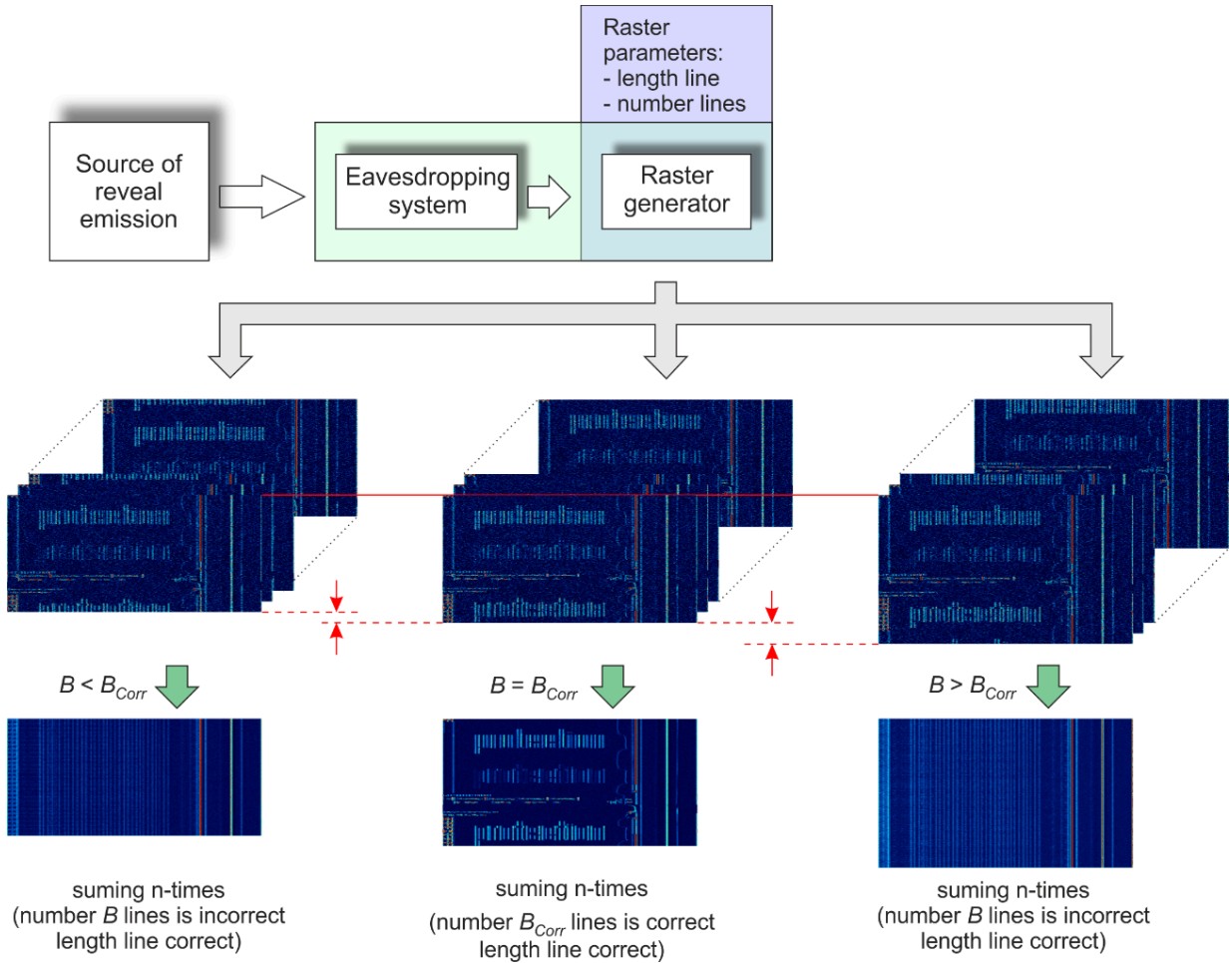

**Figure 4.** The working scheme of coherent summing of images in electromagnetic infiltration process.

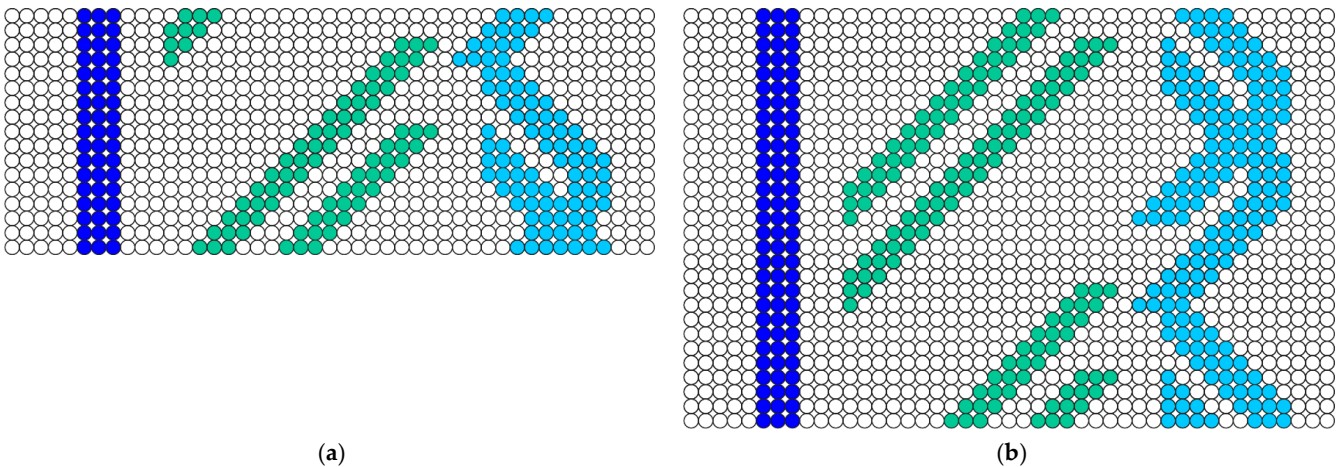

(**a**)

(**b**)

**Figure 5.** Illustrating the influence of a wrongly determined number lines of image on the result of coherent summation (a summation of two first images): (**a**) for a small number lines of image, (**b**) for a large number lines of image.

The work on images summed up several times is proposed, assuming the use of a previously determined, correct value of the $d_\Delta$ line length of the reconstructed image ($\Delta$—accuracy of the image line length estimation, where $\Delta = 10^0$, $10^{-1}$, $10^{-2}$, $10^{-3}$, $10^{-4}$, $10^{-5}$). Then, the maximum contrast should be achieved for the number $B_{Corr}$ line of the

image corresponding to the number of the original image (Figure 6). This approach causes the pixel amplitude values to change by averaging them, and the maximum average value should be achieved for the correct number of image lines.

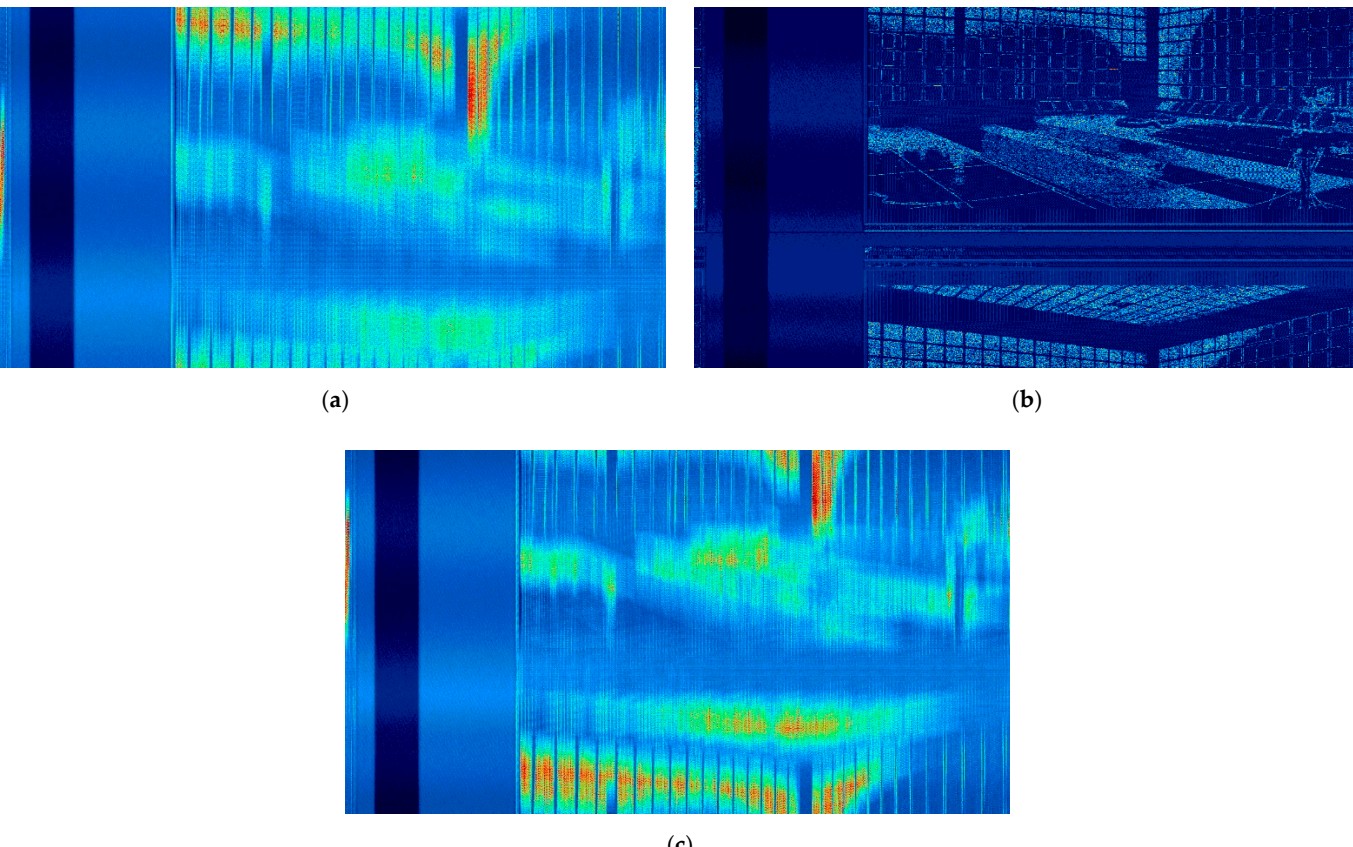

(**a**)　　　　　　　　　　　　　　　　　　　　　　　　　　　　(**b**)

(**c**)

**Figure 6.** Examples of incorrectly determined number of lines of the reconstructed images: (**a**) too few lines, (**b**) correct number of lines, (**c**) too many lines—30-fold summation of the reconstructed image for the revealing emission signal measured at the frequency $f_o = 1334$ MHz, band reception $BW = 50$ MHz, primary image displayed in the mode of $1280 \times 1024/60$ Hz, DVI standard, image in greyscale.

Additionally, three other measures have been proposed to enable the determination of the number of lines of the reproduced image. These are measures based on the methods of determining the correct length d of the image line. Nevertheless, in the case described in [28], the basis was the reconstructed images that were not processed by the use of the coherent summation algorithm. The criterion for determining the correct number $B_{Corr}$ of the image line is also the maximization of the value of the proposed measure.

Conventional methods, which are used to assess an image's contrast, can be applied to determine the number lines in a reconstructed image. These methods sometime require counting a lot of parameters before the achievement of purpose connected with the value of contrast. Therefore, simple methods are needed to determine the number lines in a reconstructed image. The simple method should not require a count of the square of amplitude pixels or a multiple sum, average value of amplitude pixels. Such operations lengthen calculation time, which is very important in the process of electromagnetic infiltration. Simultaneously, such a method has to be effective and resistant to disturbances in the determination process of number lines of reconstructed images (such images are characterized by a very low level of quality and include a lot of graphic elements that aren't valuable data from the viewpoint of the eavesdropping process).

## 2. Conditions of Conducted Tests

### 2.1. Test Images

Analyses concerning the possibility of using typical measures of contrast assessment and the methods proposed by the authors of the article were carried out on the basis of the test images presented in Figure 7. The selection of the images was based on the research experience related to the assessment of devices intended for processing classified information, which may be a source of undesirable electromagnetic emissions, and the analyses carried out in [34–37].

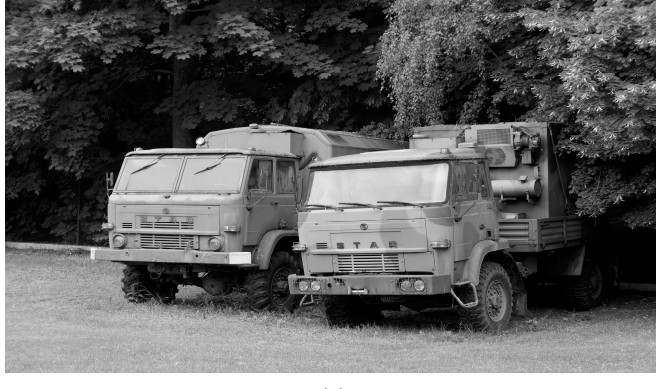

**(a)**　**(b)**

**(c)**　**(d)**

**Figure 7.** Test images used in the analysis of the effectiveness of the proposed measures in the process of determining the correct number of lines of the reproduced image: (**a**) a photo showing two vehicles (HDMI standard), (**b**) a three-column text (HDMI standard), (**c**) three words "protection" written in secure font (VGA standard), (**d**) menu of multifunctional device.

Images contain different data structures in the form of text, photos, and the menu that is provided to the user of MFPs. In the case of the first two types of images, the computer set worked with the use of the VGA and HDMI graphic standards.

### 2.2. Test Conditions

The tests were carried out in an anechoic chamber (Figure 8). The measurement system FSWT26 receiver from Rohde & Schwarz with a set of measurement antennas (a vertical active rod antenna (100 Hz up to 50 MHz), a biconical active antenna (20 MHz up to 200 MHz), and a dipole active antenna (200 MHz up to 1000 MHz)) were used in the tests. The distance between antenna and the PDA-1000 8-bit analogue-to-digital converter card was used to sample of the revealing emission signals (Signatec PDA100 Scope Application software, version 1.19). The card offers a signal sampling rate of 1 GS/s. The sampling rate can be reduced by using a card clock frequency division in the range from 2 to 1024.

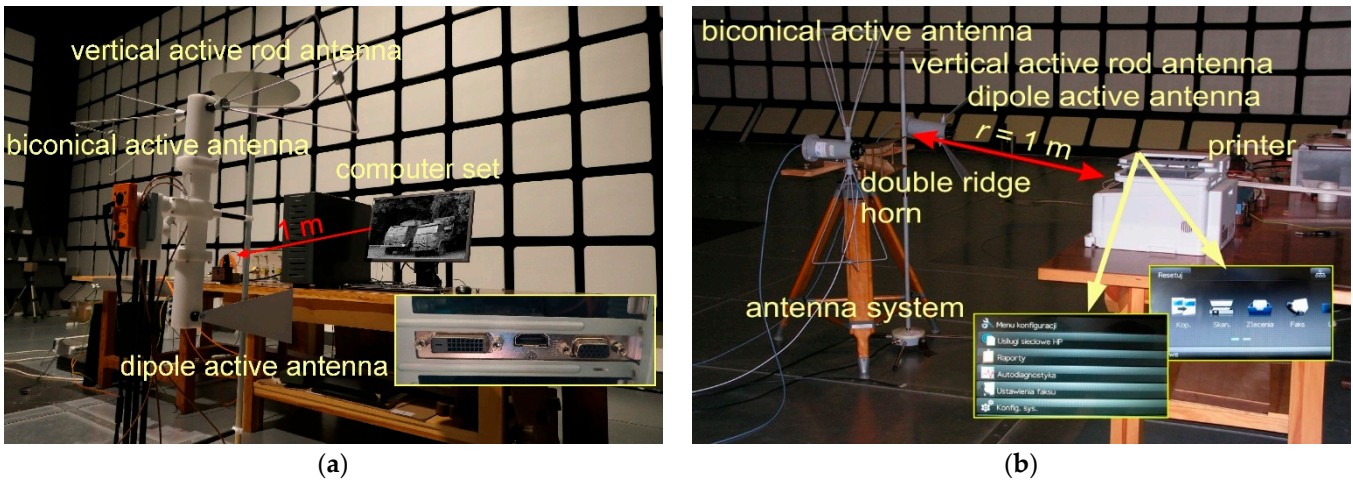

(**a**)          (**b**)

**Figure 8.** Measuring systems for three different sources of revealing emissions: (**a**) HDMI and DVI standards, (**b**) display of laser printers.

The tests were carried out using a computer set (Figure 8a) and a laser printer—HP Color Laser Jet M477fdn (Figure 8b). During testing of revealing emissions from a computer set, the monitor was operated at $1280 \times 1024/60$ Hz (HDMI standard) or $1024 \times 768/60$ Hz (VGA standard). The printer was not connected to the computer. In case of text data, black-letter characters were displayed on a white background. Images were presented in greyscale.

### 2.3. Algorithm of the Determining the Correct Number Lines of Image

Details determination of the number lines of the reconstructed image are described in the form of the algorithm shown in Figure 9.

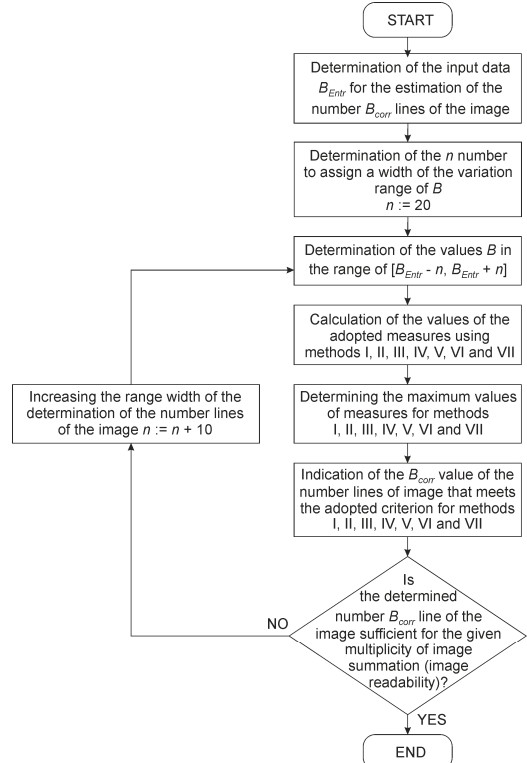

**Figure 9.** An algorithm for determining the correct number lines of a reconstructed image.

## 3. Methods of Determining of Image Number Lines

### 3.1. Introduction

Determination of the correct number $B_{Corr}$ lines of the reconstructed image is performed based on this image. This may be a reconstructed image, or an image subjected to a preselected image processing algorithm. In the case of a reconstructed image that has not undergone previous transformations causing changes in the pixel amplitude values, the use of contrast evaluation methods makes it impossible to determine the correct number of lines forming the image. Changing the number of lines only affects the data content in the image (Figure 10)—the data may be incomplete (too few lines, Figure 10a) or the data may be duplicated (too many lines, Figure 10c). Hence, the image used to determine the number of lines must undergo appropriate processing, for which the values of pixel amplitudes change as a function of changes in the number of lines in the image.

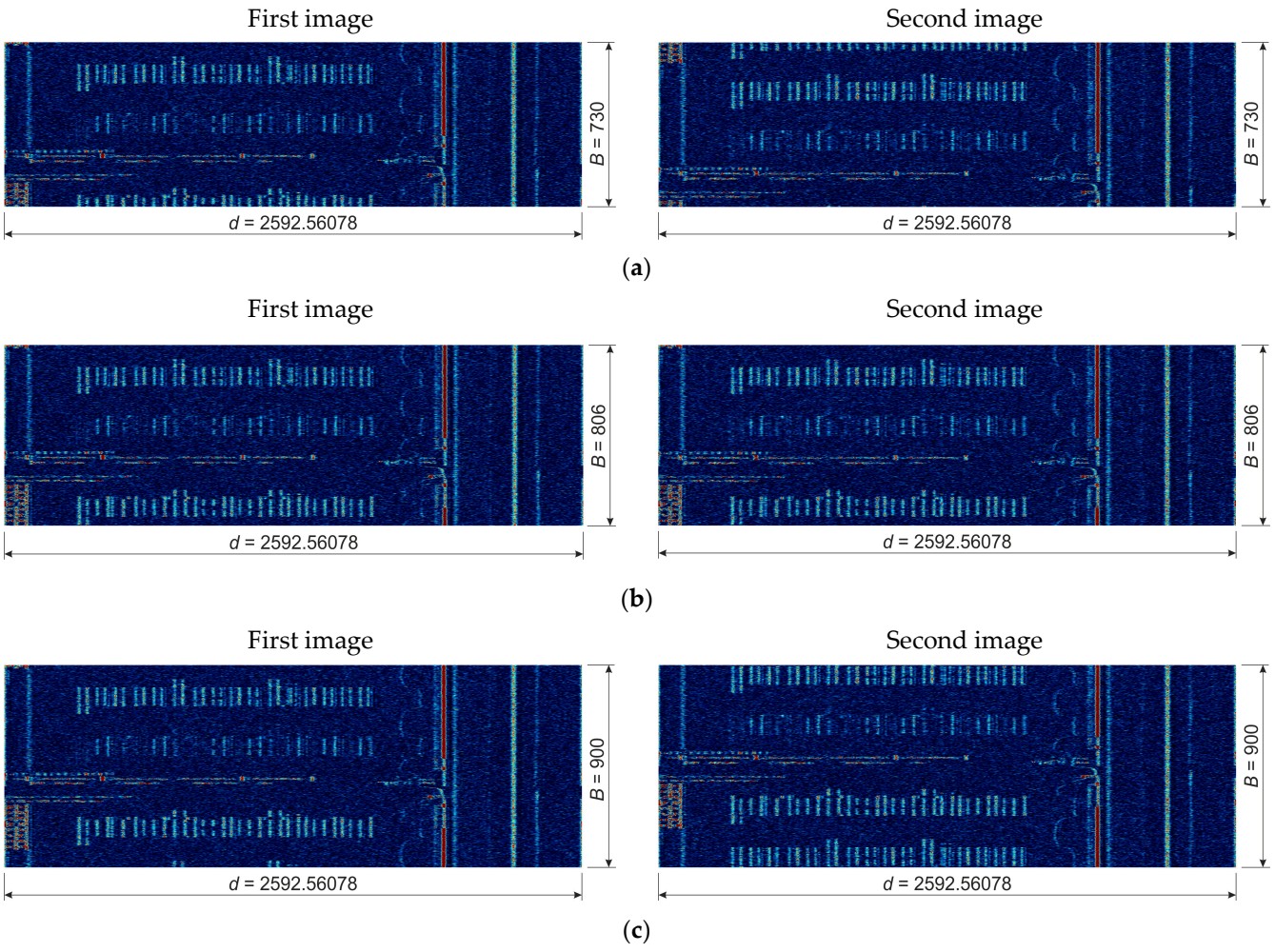

**Figure 10.** Image reconstructed (two consecutive realizations of the image) from the recorded revealing signal for: (**a**) too few image lines, $B = 730$; (**b**) correct number of image lines, $B_{Corr} = 806$; (**c**) too many image lines, $B = 900$ .

The maximum value of the adopted image quality assessment measure unambiguously indicates the correct number of $B_{Corr}$ lines for the reconstructed image.

The correct number lines of image is determined primarily from the point of view of the possibility of efficient coherent summation of tens of realizations of the reconstructed image. Too many or too few lines mean that the coherent summation process does not sharpen the data contained in the image, which is the goal of the process, but introduces additional blurring. In particular, it is noticeable when the sum of images is large. Hence,

the basis for the analysis of the usefulness of the methods of contrast assessment and the methods proposed by the authors of the article were images that are the sum of tens realizations of the same reconstructed image, for a previously determined length d of the image line.

### 3.2. Methods of Evaluating Contrast of Reconstructed Images

In order to determine the number lines of reconstructed images from the revealing emission signals, four basic methods were used to evaluate image contrast. At this point, it should be emphasized that these methods effectively determine the level of contrast of images of satisfactory quality, e.g., from data display systems in the form of photos [38,39]. The data obtained in the process of electromagnetic infiltration are recreated and presented in the form of images. The images obtained in this way are very often highly noisy, containing graphic elements that strongly contrast with the background and have no relation to the processed information [11,17,19,35]. This largely hinders the analysis of such images and the possibility of direct use of typical image processing methods [40–42].

The input data of the reconstructed image line $B_{Corr}$ algorithm are the $B_{Entr}$ quantities, which are determined at the preliminary stage of analyzing the time course of the amplitude variation of the revealing emission signal. The authors' experience shows that the $B_{Entr}$ estimation's accuracy is $\pm 10$, hence the need to clarify this value, which allows for effective processing of the obtained image in the process of tens of coherent summations.

The algorithm for determining the correct number lines of image assumes carrying out appropriate calculations of the values of the adopted measures for the number of lines:

$$B = B_{Entr} \pm n, \tag{1}$$

where $n = 0, 1, 2, \ldots, 20$.

In this way, 41 values are obtained:

$B = B_{Entr} - 20,$
$B = B_{Entr} - 19,$
$B = B_{Entr} - 18,$
$\ldots,$
$B = B_{Entr},$
$\ldots,$
$B = B_{Entr} + 18,$
$B = B_{Entr} + 19,$
$B = B_{Entr} + 20,$

selected measures that allow for the selection of the $B_{Corr}$ value that meets the adopted criterion, which is the maximum value. It corresponds to the correct number of lines of the reconstructed image.

### 3.2.1. Contrast Evaluation Based on the Value of the Average Amplitude of Pixels of the Reconstructed Image—Method I

$$Contrast_{I\_Un}(B) = \frac{Contrast_I(B)}{maximum_I}, \tag{2}$$

where

$$Contrast_I(B) = \frac{l_{B\_max}(B) - l_{B\_min}(B)}{\overline{l_B}(B)}, \tag{3}$$

$$maximum_I = \max_B (Contrast_I(B)), \tag{4}$$

$$\overline{l_B} = \frac{1}{B \cdot M} \sum_{m=0}^{M-1} \sum_{b=0}^{B-1} l_B(b, m), \tag{5}$$

$$l_{B\_max}(B) = \max_{b,m}(l_B(b,m)), \tag{6}$$

$$l_{B\_min}(B) = \min_{b,m}(l_B(b,m)), \tag{7}$$

$M$—columns number of reconstructed image;

$m$—number of columns of reconstructed image ($m = 0, 1, 2, \ldots, M - 1$);

$B$—rows number (lines) of reconstructed image calculated according to (1);

$b$—number of rows (lines) of reconstructed image ($b = 0, 1, 2, \ldots, B - 1$);

$l_B(b,m)$—value of image pixel amplitude for coordinates $(b,m)$;

$l_{B\_max}(B)$—the maximum value of the image pixel amplitude for the number $B$ line;

$l_{B\_min}(B)$—the minimum value of the image pixel amplitude for the number $B$ line.

The maximum value determined by (8) is assumed as the criterion for determining the correct $B_{Corr}$ number of the image line:

$$Contrast\_max_{I\_Un} = \max_{B}(Contrast_{I\_Un}(B)) \rightarrow B_{Corr}. \tag{8}$$

### 3.2.2. Contrast Evaluation Based on the Maximum and Minimum Values of the Amplitude of Pixels of the Reconstructed Image—Method II

$$Contrast_{II\_Un}(B) = \frac{Contrast_{II}(B)}{maximum_{II}}, \tag{9}$$

where

$$Contrast_{II}(B) = \frac{l_{B\_max}(B) - l_{B\_min}(B)}{l_{B\_max}(B) + l_{B\_min}(B)}, \tag{10}$$

$$maximum_{II} = \max_{B}(Contrast_{II}(B)), \tag{11}$$

The maximum value determined by (12) is assumed as the criterion for determining the correct $B_{Corr}$ number of the image line:

$$Contrast\_max_{II\_Un} = \max_{B}(Contrast_{II\_Un}(B)) \rightarrow B_{Corr}. \tag{12}$$

### 3.2.3. Contrast Evaluation Based on the Sum of the Differences between Adjacent Image Pixels—Method III

$$Contrast_{III\_Un}(B) = \frac{Contrast_{III}(B)}{maximum_{III}}, \tag{13}$$

where:

$$
\begin{aligned}
Contrast_{III}(B) = \quad & \frac{1}{B \cdot M \cdot 255^2} \left( \sum_{m=0}^{M-1}\sum_{b=0}^{B-2}(l_B(b,m) - l_B(b+1,m))^2 \right. \\
& + \sum_{m=0}^{M-2}\sum_{b=0}^{B-1}(l_B(b,m) - l_B(b,m+1))^2 \\
& + \sum_{m=0}^{M-2}\sum_{b=0}^{B-2}(l_B(b,m) - l_B(b+1,m+1))^2 \\
& \left. + \sum_{m=0}^{M-1}\sum_{b=0}^{B-2}(l_B(b,m) - l_B(b+1,m-1))^2 \right)
\end{aligned}
\tag{14}
$$

$$maximum_{III} = \max_{B}(Contrast_{III}(B)). \tag{15}$$

The maximum value determined by (16) is assumed as the criterion for determining the correct $B_{Corr}$ number of the image line:

$$Contrast\_max_{III\_Un} = \max_{B}(Contrast_{III\_Un}(B)) \rightarrow B_{Corr}. \tag{16}$$

### 3.2.4. Contrast Evaluation Based on the Variance of the Grey Levels of the Reconstructed Image—Method IV

$$Contrast_{IV\_Un}(B) = \frac{Contrast_{IV}(B)}{maximum_{IV}}, \tag{17}$$

where

$$Contrast_{IV}(B) = \frac{4}{B \cdot M \cdot 255^2} \sum_{m=0}^{M-1}\sum_{b=0}^{B-1} \left[ l_B(b,m) - \overline{l_B}(B) \right]^2, \tag{18}$$

$$maximum_{IV} = \max_{B}(Contrast_{IV}(B)). \tag{19}$$

The maximum value determined by (20) is assumed as the criterion for determining the correct $B_{Corr}$ number of the image line:

$$Contrast\_max_{IV\_Un} = \max_{B}(Contrast_{IV\_Un}(B)) \rightarrow B_{Corr}. \tag{20}$$

### 3.3. Methods Proposed by Authors

The methods proposed by the authors of the article are based only on the values of the maximum and minimum pixel amplitudes and their differences. The mean values and variances of the amplitudes of pixels composing the analysed image are not calculated. Therefore, taking into account the time-consuming computation for images of large sizes, e.g., for sources of unwanted emission in the form of graphic paths of laser printers or monitors operating in higher graphic modes, these methods may be effective in electromagnetic infiltration processes.

3.3.1. The Maximum Value of the Difference between the Maximum and Minimum Pixel Amplitude Sums Calculated for Each Vertical Line of the Reconstructed Image—Method V

Method V is not based directly on the maximum and minimum amplitude values of the pixels building the reconstructed image. The respective maximum $Maximum_{Met\_V}(B)$ and minimum $Minimum_{Met\_V}(B)$ values are calculated for the sums $Sum_{Met\_V}(B,m)$ of pixel amplitude values calculated for each column of the analysed image according to formulas:

$$Sum_{Met\_V}(B,m) = \sum_{b=0}^{B-1} l_B(b,m), \tag{21}$$

$$Maximum_{Met\_V}(B) = \max_{m}(Sum_{Met\_V}(B,m)), \tag{22}$$

$$Minimum_{Met\_V}(B) = \min_{m}(Sum_{Met\_V}(B,m)). \tag{23}$$

Then, according to the adopted algorithm, the differences $Dif_{Met\_V}(B)$ between the maximum value $Maximum_{Met\_V}(B)$ and the minimum $Minimum_{Met\_V}(B)$ of sums $Sum_{Met\_V}(B,m)$ are calculated, according to the formula

$$Dif_{Met\_V}(B) = Maximum_{Met\_V}(B) - Minimum_{Met_V}(B). \tag{24}$$

The next stage of the procedure requires the determination of the maximum value $Maximum_{Dif_{Met\_V}}$:

$$Maximum_{Dif_{Met\_V}} = \max_{B}(Dif_{Met\_VI}(B)), \tag{25}$$

which allows us to calculate normalized values:

$$Dif_{Met\_V\_Un}(B) = \frac{Dif_{Met\_V}(B)}{Maximum_{Dif_{Met\_V}}}. \tag{26}$$

The maximum value determined by (27) is assumed as the criterion for determining the correct $B_{Corr}$ number of the image line:

$$Dif\_max_{Met\_V\_Un} = \max_B(Dif_{Met\_V\_Un}(B)) \rightarrow B_{Corr}. \tag{27}$$

3.3.2. The Minimum Value of the Sum of the Differences of the Maximum and Minimum Amplitudes Calculated for Individual Vertical Lines of the Reconstructed Image—Method VI

Method VI requires the calculation of the maximum $Maximum_{Met\_VI}(B, m)$ and minimum $Minimum_{Met\_VI}(B, m)$ values of the pixel amplitude for each column $m$ of the reconstructed image according to the formula

$$Maximum_{Met\_VI}(B, m) = \max_b(l_B(b, m)), \tag{28}$$

$$Minimum_{Met\_VI}(B, m) = \min_b(l_B(b, m)). \tag{29}$$

Next, a sum of differences $Maximum_{Met\_VI}(B, m) - Minimum_{Met\_VI}(B, m)$ is calculated:

$$Sum_{Met\_VI}(B) = \sum_{m=0}^{M-1}(Maximum_{Met\_VI}(B, m) - Minimum_{Met\_VI}(B, m)), \tag{30}$$

which is calculated independently for each value of B. In the next step, the maximum value of $Sum_{Met\_V}(B)$ is determined:

$$Maximum\_Sum_{Met\_VI} = \max_B(Sum_{Met\_VI}(B)) \tag{31}$$

allowing for the calculation of normalized values:

$$Sum_{Met\_VI\_Un}(B) = \frac{Sum_{Met\_VI}(B)}{Maximum\_Sum_{Met\_VI}}. \tag{32}$$

The maximum value determined by (33) is assumed as the criterion for determining the correct $B_{Corr}$ number of the image line:

$$Sum\_max_{Met\_V\_Un} = \max_B\left(\frac{Sum_{Met\_V}(B)}{Maximum\_Sum_{Met\_V}}\right) \rightarrow B_{Corr}. \tag{33}$$

3.3.3. The Minimum Value of the Sum of the Maximum Pixel Amplitudes Calculated for the Individual Vertical Lines of the Reconstructed Image—Method VII

Method VII is similar to method VI, requiring only the calculation of the sums of the maximum values $Maximum_{Met\_VII}(B, m)$ image pixel amplitudes for the $B$ values determined for each column $m$ of the reconstructed image according to the formula:

$$Sum_{Met\_VII}(B) = \sum_{m=0}^{M-1} Maximum_{Met\_VII}(B, m). \tag{34}$$

Further stages of the procedure are the same as in the case of the VI method and require the determination of the maximum value of $Maximum\_Sum_{Met\_VII}$:

$$Maximum\_Sum_{Met\_VII} = \max_{B}(Sum_{Met\_VII}(B)), \tag{35}$$

which allows us to calculate normalized values:

$$Sum_{Met\_VII\_Un}(B) = \frac{Sum_{Met\_VII}(B)}{Maximum\_Sum_{Met\_VII}}. \tag{36}$$

The maximum value determined by (37) is assumed as the criterion for determining the correct $B_{Corr}$ number of the image line:

$$Sum\_max_{Met\_VII\_Un} = \max_{B}(Sum_{Met\_VII\_Un}(B)) \rightarrow B_{Corr}. \tag{37}$$

## 4. Test Results

The input datam in the conducted analyses was the $B_{Entr}$ value, which is the number of image lines estimated on the basis of preliminary analyses of the variability of the amplitude values of the undesirable emission signal, which is the basis of the screening process. As shown in Figure 3, the data contained in the reconstructed image are periodically repeated, which facilitates the preliminary determination of the number of image lines. Its more precise determination requires the use of an algorithm based on the proposed measures and criteria. The final verification of the correctness determination of the number of image lines is carried out on the basis of visual assessment, i.e., the readability of the data contained in the image after using coherent summation of several dozen image realizations. The summation process is intended to improve image quality.

### 4.1. HDMI Standard as a Source of Reveal Emissions—Sample Images

4.1.1. Primary Image in the Form of the Photo Presented in Figure 7a ($B_{Entr} = 1069$, $d = 195,364,892$)

Figure 11 (Table 1) shows the changes in the values of the adopted measures (methods) calculated as a function of the number lines of the image reconstructed from the recorded revealing emission signal. The criterion for which the correct number of image lines is indicated is the maximum value of the calculated measure (method, Figure 12).

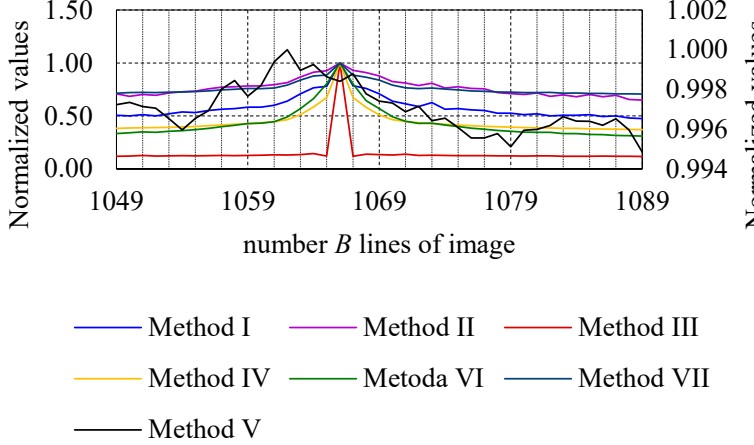

**Figure 11.** Normalized values (in relation to the maximum value) of the variability of measures as a function of the number of image lines, supporting the determination of the correct number of lines of the image reproduced for the source image in the form of a picture from Figure 7a—HDMI graphic standard, computer monitor operating mode $1280 \times 1024/60$ Hz, frequency of the reveal emission signal $f_o = 1334$ MHz, reception bandwidth $BW = 50$ MHz, correct number of lines $B_{Corr} = 1066$.

**Table 1.** Normalized values (in relation to the maximum value) of the variability of measures as a function of the number of image lines, supporting the determination of the correct number of lines of the image reproduced for the source image in the form of a picture from Figure 7a.

| Number Lines | Method I | Method II | Method III | Method IV | Method V | Method VI | Method VII |
|---|---|---|---|---|---|---|---|
| 1049 | 0.48112 | 0.66368 | 0.12038 | 0.37683 | 0.99667 | 0.31955 | 0.71072 |
| 1050 | 0.48110 | 0.66368 | 0.12021 | 0.37914 | 0.99715 | 0.32650 | 0.71337 |
| 1051 | 0.49375 | 0.68114 | 0.12035 | 0.38169 | 0.99745 | 0.33138 | 0.71521 |
| 1052 | 0.50641 | 0.70756 | 0.11987 | 0.38257 | 0.99724 | 0.33263 | 0.71511 |
| 1053 | 0.50008 | 0.68553 | 0.12118 | 0.38610 | 0.99735 | 0.34176 | 0.71963 |
| 1054 | 0.51276 | 0.70289 | 0.12727 | 0.38881 | 0.99713 | 0.34896 | 0.72230 |
| 1055 | 0.50012 | 0.69427 | 0.12123 | 0.38966 | 0.99706 | 0.34518 | 0.71935 |
| 1056 | 0.51913 | 0.71607 | 0.12294 | 0.39326 | 0.99653 | 0.35504 | 0.72366 |
| 1057 | 0.53815 | 0.72844 | 0.12468 | 0.39653 | 0.99597 | 0.36058 | 0.72539 |
| 1058 | 0.53175 | 0.73354 | 0.12395 | 0.40099 | 0.99654 | 0.36990 | 0.72964 |
| 1059 | 0.55056 | 0.75496 | 0.12560 | 0.40755 | 0.99693 | 0.38265 | 0.73666 |
| 1060 | 0.56307 | 0.77231 | 0.12623 | 0.41467 | 0.99802 | 0.39878 | 0.74565 |
| 1061 | 0.56945 | 0.77611 | 0.12566 | 0.42114 | 0.99845 | 0.41260 | 0.75270 |
| 1062 | 0.58229 | 0.78356 | 0.12731 | 0.42737 | 0.99767 | 0.42734 | 0.75961 |
| 1063 | 0.58216 | 0.78356 | 0.12921 | 0.43537 | 0.99821 | 0.43099 | 0.75837 |
| 1064 | 0.60099 | 0.79440 | 0.13341 | 0.44595 | 0.99939 | 0.44564 | 0.76384 |
| 1065 | 0.63890 | 0.81494 | 0.13197 | 0.46566 | 1.00000 | 0.49005 | 0.79045 |
| 1066 | 0.70862 | 0.86816 | 0.13563 | 0.51140 | 0.99897 | 0.56892 | 0.83785 |
| 1067 | 0.76568 | 0.91229 | 0.14481 | 0.58404 | 0.99926 | 0.66339 | 0.87759 |
| 1068 | 0.77841 | 0.92737 | 0.12076 | 0.67389 | 0.99864 | 0.80281 | 0.88614 |
| 1069 | 1.00000 | 1.00000 | 1.00000 | 1.00000 | 0.99840 | 1.00000 | 1.00000 |
| 1070 | 0.78485 | 0.92983 | 0.11941 | 0.67282 | 0.99878 | 0.80118 | 0.88511 |
| 1071 | 0.75955 | 0.90972 | 0.13853 | 0.58123 | 0.99778 | 0.64458 | 0.86525 |
| 1072 | 0.70892 | 0.87802 | 0.13447 | 0.51034 | 0.99741 | 0.56897 | 0.83815 |
| 1073 | 0.63931 | 0.82458 | 0.13166 | 0.46514 | 0.99732 | 0.49238 | 0.79136 |
| 1074 | 0.61399 | 0.81112 | 0.13834 | 0.44651 | 0.99688 | 0.44840 | 0.76632 |
| 1075 | 0.58869 | 0.78722 | 0.12767 | 0.43508 | 0.99715 | 0.42876 | 0.75767 |
| 1076 | 0.62670 | 0.80825 | 0.12894 | 0.42835 | 0.99642 | 0.43172 | 0.76325 |
| 1077 | 0.56340 | 0.76272 | 0.12668 | 0.42104 | 0.99655 | 0.41378 | 0.75347 |
| 1078 | 0.56973 | 0.77611 | 0.12589 | 0.41452 | 0.99608 | 0.40084 | 0.74680 |
| 1079 | 0.55709 | 0.75886 | 0.12571 | 0.40814 | 0.99556 | 0.38421 | 0.73693 |
| 1080 | 0.55076 | 0.75496 | 0.12554 | 0.40280 | 0.99555 | 0.37415 | 0.73254 |
| 1081 | 0.52544 | 0.72025 | 0.12342 | 0.39780 | 0.99577 | 0.36193 | 0.72628 |
| 1082 | 0.52545 | 0.71130 | 0.12327 | 0.39436 | 0.99512 | 0.35500 | 0.72396 |
| 1083 | 0.51278 | 0.70289 | 0.12137 | 0.39026 | 0.99595 | 0.34622 | 0.72014 |
| 1084 | 0.51911 | 0.71607 | 0.12295 | 0.38815 | 0.99599 | 0.34502 | 0.72043 |
| 1085 | 0.50009 | 0.68553 | 0.12346 | 0.38674 | 0.99619 | 0.34480 | 0.72162 |
| 1086 | 0.50642 | 0.69861 | 0.11976 | 0.38258 | 0.99663 | 0.33250 | 0.71538 |
| 1087 | 0.50640 | 0.68136 | 0.12013 | 0.38166 | 0.99640 | 0.33270 | 0.71585 |
| 1088 | 0.51273 | 0.70289 | 0.11969 | 0.37905 | 0.99640 | 0.32642 | 0.71335 |
| 1089 | 0.49375 | 0.68114 | 0.12204 | 0.37712 | 0.99619 | 0.32428 | 0.71281 |

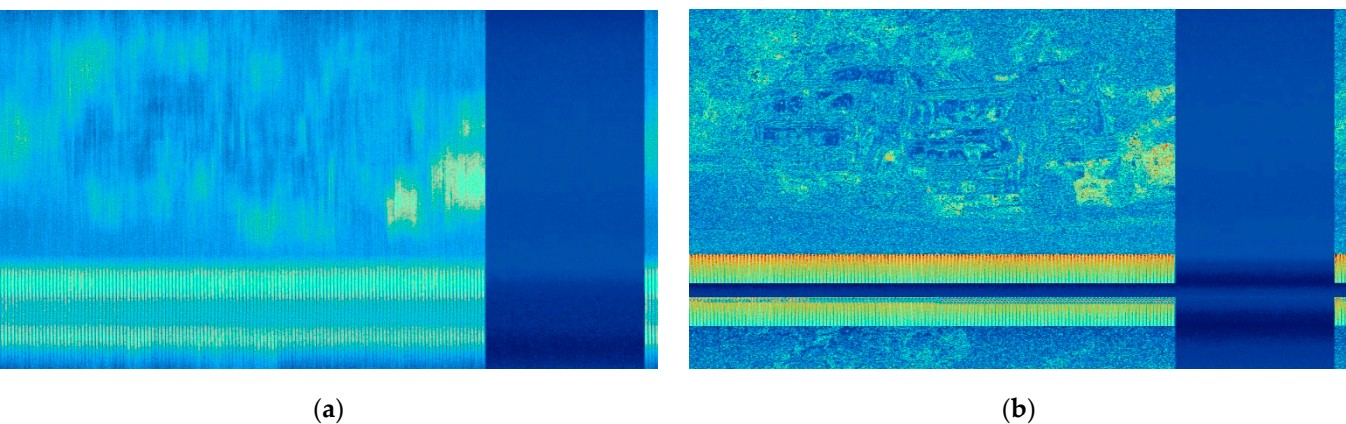

(**a**)                                                                                    (**b**)

**Figure 12.** Images reconstructed from the revealing emission signal measured at the frequency $f_o = 1334$ MHz for the number of lines of this image determined in accordance with the criterion of the maximum value of the measures presented in Figure 11: (**a**) the number of image lines determined in accordance with method V ($B = 1062$, the number of lines smaller than required), (**b**) number of image lines determined in accordance with methods I, II, III, IV, VI, and VII ($B_{Corr} = 1066$, correct number of lines).

4.1.2. Primary Image in the Form of the Text Presented in Figure 7b ($B_{Entr} = 1125$, d =741,466,826)

Figure 13 (Table 2) shows the changes in the values of the adopted measures (methods) calculated as a function of the number lines of the image reconstructed from the recorded revealing emission signal. The criterion for which the correct number of image lines is indicated is the maximum value of the calculated measure (method, Figure 14).

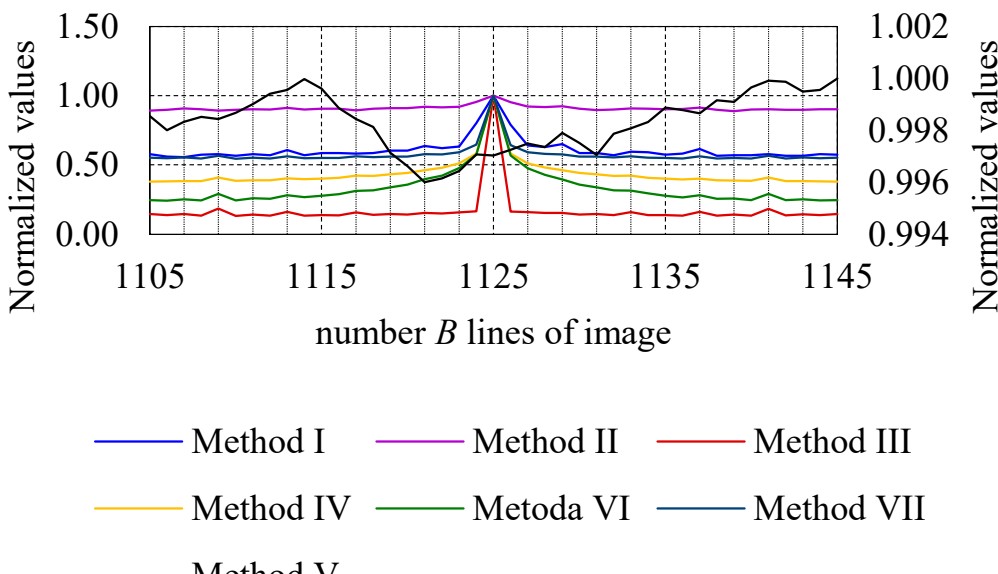

**Figure 13.** Normalized values (in relation to the maximum value) of the variability of measures as a function of the number of image lines, supporting the determination of the correct number of lines of the image reproduced for the source image in the form of a picture from Figure 7b—HDMI graphic standard, computer monitor operating mode $1280 \times 1024/60$ Hz, frequency of the reveal emission signal $f_o = 200$ MHz, reception bandwidth $BW = 100$ MHz, correct number of lines $B_{Corr} = 1125$.

**Table 2.** Normalized values (in relation to the maximum value) of the variability of measures as a function of the number of image lines, supporting the determination of the correct number of lines of the image reproduced for the source image in the form of a picture from Figure 7b.

| Number Lines | Method I | Method II | Method III | Method IV | Method V | Method VI | Method VII |
|---|---|---|---|---|---|---|---|
| 1105 | 0.57762 | 0.89215 | 0.14516 | 0.37915 | 0.99855 | 0.24556 | 0.55184 |
| 1106 | 0.56075 | 0.89722 | 0.13708 | 0.38025 | 0.99800 | 0.24318 | 0.54736 |
| 1107 | 0.55649 | 0.90676 | 0.14503 | 0.38373 | 0.99833 | 0.25174 | 0.55427 |
| 1108 | 0.57335 | 0.90127 | 0.13439 | 0.38217 | 0.99851 | 0.24458 | 0.54638 |
| 1109 | 0.57754 | 0.89215 | 0.18423 | 0.40866 | 0.99843 | 0.29020 | 0.56635 |
| 1110 | 0.56487 | 0.89858 | 0.13218 | 0.38417 | 0.99867 | 0.24414 | 0.54494 |
| 1111 | 0.57752 | 0.90258 | 0.14120 | 0.38847 | 0.99901 | 0.25906 | 0.55138 |
| 1112 | 0.56902 | 0.89993 | 0.13411 | 0.38971 | 0.99940 | 0.25478 | 0.54643 |
| 1113 | 0.60694 | 0.91139 | 0.16184 | 0.40234 | 0.99955 | 0.28113 | 0.56222 |
| 1114 | 0.56900 | 0.89993 | 0.13398 | 0.39623 | 0.99997 | 0.26640 | 0.54818 |
| 1115 | 0.58582 | 0.90517 | 0.13849 | 0.40093 | 0.99960 | 0.27724 | 0.55031 |
| 1116 | 0.58584 | 0.90517 | 0.13491 | 0.40621 | 0.99885 | 0.28810 | 0.55000 |
| 1117 | 0.58169 | 0.89350 | 0.15796 | 0.42220 | 0.99843 | 0.31262 | 0.56130 |
| 1118 | 0.58603 | 0.90517 | 0.13985 | 0.42119 | 0.99813 | 0.31692 | 0.55634 |
| 1119 | 0.60320 | 0.91018 | 0.14516 | 0.43215 | 0.99713 | 0.33761 | 0.55992 |
| 1120 | 0.60339 | 0.91018 | 0.14191 | 0.44263 | 0.99662 | 0.35726 | 0.56085 |
| 1121 | 0.63727 | 0.91953 | 0.15370 | 0.46147 | 0.99601 | 0.39707 | 0.57700 |
| 1122 | 0.62035 | 0.91496 | 0.14898 | 0.47962 | 0.99615 | 0.42268 | 0.57586 |
| 1123 | 0.63299 | 0.91841 | 0.15725 | 0.51080 | 0.99643 | 0.47718 | 0.59181 |
| 1124 | 0.80171 | 0.95558 | 0.16425 | 0.58135 | 0.99707 | 0.57332 | 0.64581 |
| 1125 | 1.00000 | 1.00000 | 1.00000 | 1.00000 | 0.99703 | 1.00000 | 1.00000 |
| 1126 | 0.78902 | 0.95326 | 0.16353 | 0.58082 | 0.99725 | 0.56847 | 0.64176 |
| 1127 | 0.64129 | 0.92065 | 0.15858 | 0.51102 | 0.99749 | 0.47550 | 0.59170 |
| 1128 | 0.62862 | 0.91727 | 0.15249 | 0.48096 | 0.99735 | 0.42839 | 0.58010 |
| 1129 | 0.64968 | 0.92284 | 0.15407 | 0.46193 | 0.99789 | 0.39447 | 0.57593 |
| 1130 | 0.58637 | 0.90517 | 0.14125 | 0.44246 | 0.99751 | 0.35709 | 0.56039 |
| 1131 | 0.58636 | 0.89483 | 0.14507 | 0.43175 | 0.99705 | 0.33826 | 0.56026 |
| 1132 | 0.56944 | 0.89993 | 0.13790 | 0.42055 | 0.99786 | 0.31653 | 0.55444 |
| 1133 | 0.59473 | 0.90770 | 0.15950 | 0.42232 | 0.99807 | 0.31523 | 0.56187 |
| 1134 | 0.59050 | 0.90645 | 0.13854 | 0.40677 | 0.99832 | 0.29376 | 0.55238 |
| 1135 | 0.57357 | 0.90127 | 0.13832 | 0.40079 | 0.99887 | 0.27779 | 0.55003 |
| 1136 | 0.58200 | 0.90389 | 0.13431 | 0.39567 | 0.99878 | 0.26565 | 0.54704 |
| 1137 | 0.61572 | 0.91378 | 0.16145 | 0.40153 | 0.99865 | 0.28136 | 0.56207 |
| 1138 | 0.56508 | 0.89858 | 0.13339 | 0.38904 | 0.99916 | 0.25475 | 0.54625 |
| 1139 | 0.56929 | 0.88941 | 0.14100 | 0.38781 | 0.99910 | 0.25648 | 0.55012 |
| 1140 | 0.56924 | 0.89993 | 0.13318 | 0.38434 | 0.99964 | 0.24651 | 0.54575 |
| 1141 | 0.57766 | 0.90258 | 0.18310 | 0.40838 | 0.99991 | 0.29033 | 0.56573 |
| 1142 | 0.56500 | 0.89858 | 0.13482 | 0.38229 | 0.99987 | 0.24598 | 0.54701 |
| 1143 | 0.56495 | 0.89858 | 0.14440 | 0.38370 | 0.99949 | 0.25112 | 0.55377 |
| 1144 | 0.57760 | 0.90258 | 0.13786 | 0.38086 | 0.99955 | 0.24335 | 0.54738 |
| 1145 | 0.57336 | 0.90127 | 0.14519 | 0.37995 | 1.00000 | 0.24640 | 0.55225 |

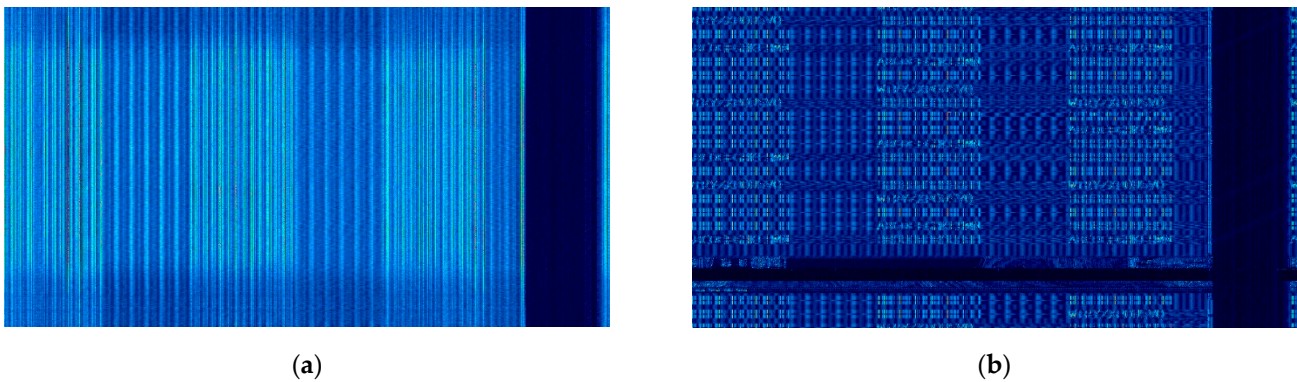

(**a**)                                                                              (**b**)

**Figure 14.** Images reconstructed from the revealing emission signal measured at the frequency $f_o = 200$ MHz for the number of lines of this image determined in accordance with the criterion of the maximum value of the measures presented in Figure 13: (**a**) the number of image lines determined in accordance with method V ($B = 1114$, the number of lines smaller than required), (**b**) number of image lines determined in accordance with methods I, II, III, IV, VI, and VII ($B_{Corr} = 1125$, correct number of lines).

*4.2. VGA Standard as a Source of Reveal Emissions—Sample Image*
Primary Image in the Form of the Text Presented in Figure 7c ($B_{Entr} = 809$, $d =$259,256,078)

Figure 15 (Table 3) shows the changes in the values of the adopted measures (methods) calculated as a function of the number lines of the image reconstructed from the recorded revealing emission signal. The criterion for which the correct number of image lines is indicated is the maximum value of the calculated measure (method, Figure 16).

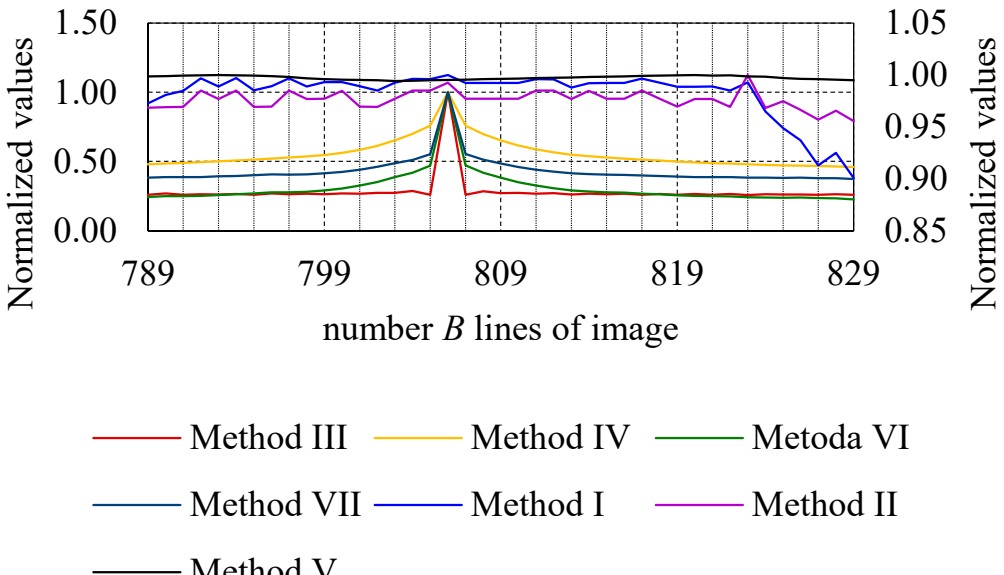

**Figure 15.** Normalized values (in relation to the maximum value) of the variability of measures as a function of the number of image lines, supporting the determination of the correct number of lines of the image reproduced for the source image in the form of a picture from Figure 7c—VGA graphic standard, computer monitor operating mode $1024 \times 768/60$ Hz, frequency of the reveal emission signal $f_o = 558$ MHz, reception bandwidth $BW = 10$ MHz, correct number of lines $B_{Corr} = 806$.

**Table 3.** Normalized values (in relation to the maximum value) of the variability of measures as a function of the number of image lines, supporting the determination of the correct number of lines of the image reproduced for the source image in the form of a picture from Figure 7c.

| Number Lines | Method I | Method II | Method III | Method IV | Method V | Method VI | Method VII |
|---|---|---|---|---|---|---|---|
| 786 | 0.93999 | 0.97422 | 0.26164 | 0.47097 | 0.99740 | 0.23701 | 0.38126 |
| 787 | 0.94417 | 0.95911 | 0.26043 | 0.47252 | 0.99787 | 0.23726 | 0.38127 |
| 788 | 0.96437 | 0.97559 | 0.26232 | 0.47590 | 0.99821 | 0.23882 | 0.38134 |
| 789 | 0.97254 | 0.96849 | 0.25853 | 0.47973 | 0.99855 | 0.24275 | 0.38286 |
| 790 | 0.98066 | 0.96898 | 0.26950 | 0.48486 | 0.99885 | 0.24939 | 0.38696 |
| 791 | 0.98467 | 0.96922 | 0.25915 | 0.48867 | 0.99928 | 0.24873 | 0.38618 |
| 792 | 0.99664 | 0.98480 | 0.26338 | 0.49419 | 0.99961 | 0.25065 | 0.38699 |
| 793 | 0.98871 | 0.97689 | 0.26055 | 0.49995 | 1.00000 | 0.25727 | 0.39192 |
| 794 | 0.99687 | 0.98480 | 0.26599 | 0.50633 | 0.99950 | 0.26268 | 0.39543 |
| 795 | 0.98504 | 0.96922 | 0.25906 | 0.51268 | 0.99933 | 0.26891 | 0.39989 |
| 796 | 0.98895 | 0.96946 | 0.26829 | 0.52101 | 0.99867 | 0.27769 | 0.40571 |
| 797 | 0.99670 | 0.98480 | 0.26246 | 0.52815 | 0.99800 | 0.27694 | 0.40438 |
| 798 | 0.98889 | 0.97689 | 0.26738 | 0.53642 | 0.99678 | 0.28178 | 0.40657 |
| 799 | 0.99298 | 0.97710 | 0.26344 | 0.54652 | 0.99590 | 0.29121 | 0.41451 |
| 800 | 0.99302 | 0.98462 | 0.26974 | 0.56238 | 0.99541 | 0.30410 | 0.42369 |
| 801 | 0.98905 | 0.96946 | 0.26764 | 0.58427 | 0.99517 | 0.32538 | 0.44039 |
| 802 | 0.98495 | 0.96922 | 0.27374 | 0.61365 | 0.99477 | 0.35233 | 0.46085 |
| 803 | 0.99260 | 0.97710 | 0.27272 | 0.65206 | 0.99423 | 0.38692 | 0.48712 |
| 804 | 0.99624 | 0.98480 | 0.28593 | 0.69870 | 0.99429 | 0.41795 | 0.51159 |
| 805 | 0.99599 | 0.98480 | 0.25964 | 0.75836 | 0.99476 | 0.46916 | 0.55245 |
| 806 | 1.00000 | 0.99256 | 1.00000 | 1.00000 | 0.99502 | 1.00000 | 1.00000 |
| 807 | 0.99208 | 0.97710 | 0.25951 | 0.75794 | 0.99526 | 0.47048 | 0.55397 |
| 808 | 0.99224 | 0.97710 | 0.28464 | 0.69811 | 0.99581 | 0.41896 | 0.51203 |
| 809 | 0.99234 | 0.97710 | 0.27025 | 0.65184 | 0.99613 | 0.38386 | 0.48466 |
| 810 | 0.99217 | 0.97710 | 0.27400 | 0.61508 | 0.99651 | 0.34956 | 0.45882 |
| 811 | 0.99596 | 0.98480 | 0.26738 | 0.58638 | 0.99687 | 0.32565 | 0.44042 |
| 812 | 0.99572 | 0.98480 | 0.27156 | 0.56520 | 0.99729 | 0.30617 | 0.42559 |
| 813 | 0.98782 | 0.97689 | 0.26211 | 0.54867 | 0.99753 | 0.29101 | 0.41444 |
| 814 | 0.99198 | 0.98462 | 0.26691 | 0.53803 | 0.99794 | 0.28382 | 0.40821 |
| 815 | 0.99215 | 0.97710 | 0.26269 | 0.52874 | 0.99835 | 0.27632 | 0.40388 |
| 816 | 0.99234 | 0.97710 | 0.26782 | 0.52114 | 0.99861 | 0.27423 | 0.40329 |
| 817 | 0.99645 | 0.98480 | 0.25994 | 0.51272 | 0.99890 | 0.26746 | 0.39883 |
| 818 | 0.99245 | 0.97710 | 0.26489 | 0.50592 | 0.99929 | 0.26227 | 0.39476 |
| 819 | 0.98845 | 0.96946 | 0.25954 | 0.49942 | 0.99955 | 0.25620 | 0.39169 |
| 820 | 0.98856 | 0.97689 | 0.26432 | 0.49381 | 0.99993 | 0.25080 | 0.38745 |
| 821 | 0.98877 | 0.97689 | 0.25971 | 0.48806 | 0.99920 | 0.24887 | 0.38690 |
| 822 | 0.98492 | 0.96922 | 0.26583 | 0.48423 | 0.99943 | 0.24802 | 0.38685 |
| 823 | 0.99273 | 1.00000 | 0.25788 | 0.47910 | 0.99857 | 0.24196 | 0.38314 |
| 824 | 0.96461 | 0.96799 | 0.26365 | 0.47488 | 0.99815 | 0.24043 | 0.38228 |
| 825 | 0.94880 | 0.97468 | 0.26036 | 0.47078 | 0.99704 | 0.23810 | 0.38149 |
| 826 | 0.93688 | 0.96619 | 0.26205 | 0.46914 | 0.99625 | 0.23997 | 0.38255 |

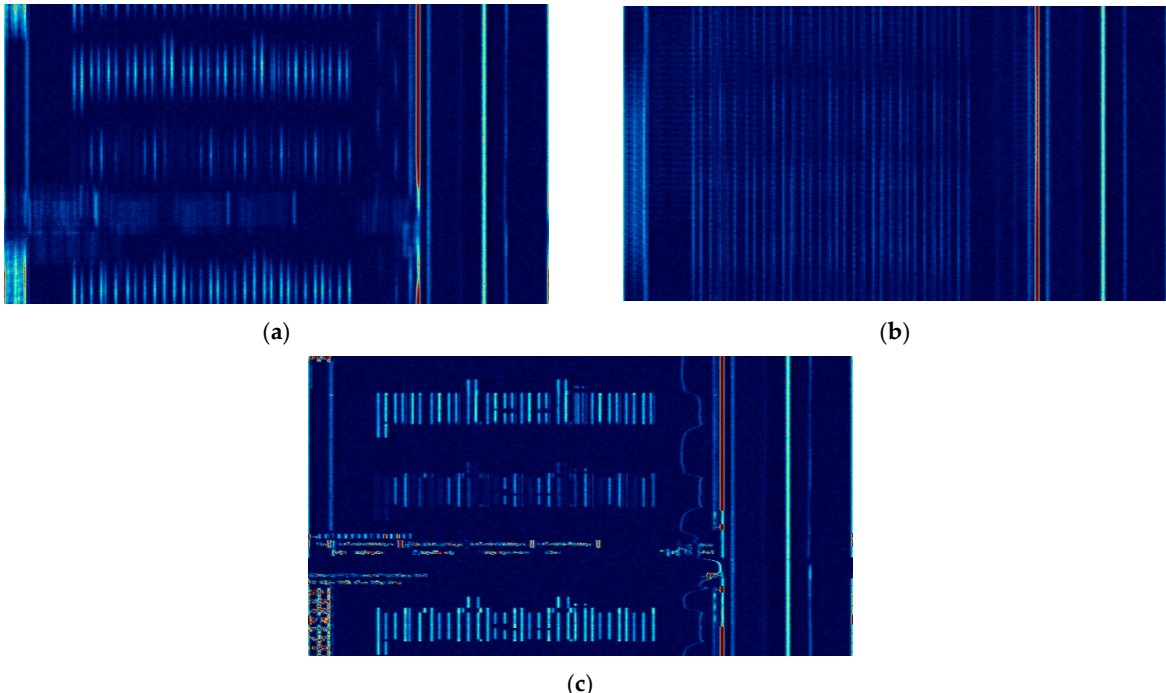

**Figure 16.** Images reconstructed from the revealing emission signal measured at the frequency $f_o = 558$ MHz, for the number of lines of this image determined in accordance with the criterion of the maximum value of the measures presented in Figure 15: (**a**) the number of image lines determined in accordance with method V ($B = 803$, the number of lines smaller than required), (**b**) number of image lines determined in accordance with method II ($B = 823$, the number of lines bigger than required), (**c**) number of image lines determined in accordance with method I, III, IV, VI, and VII ($B_{Corr} = 806$, correct number of lines).

*4.3. Display of Multifunctional Device as a Source of Reveal Emissions—Sample Image*
Primary Image in the Form of Menu Presented in Figure 7d ($B_{Entr} = 294$, $d = 93,127,351$)

Figure 17 (Table 4) shows the changes in the values of the adopted measures (methods) calculated as a function of the number lines of the image reconstructed from the recorded revealing emission signal. The criterion for which the correct number of image lines is indicated is the maximum value of the calculated measure (method, Figure 18).

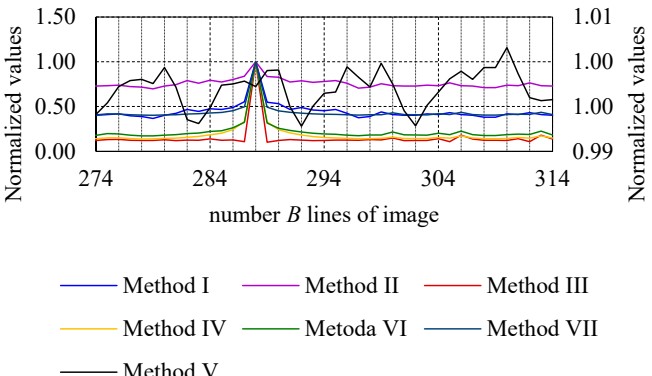

**Figure 17.** Normalized values (in relation to the maximum value) of the variability of measures as a function of the number of image lines, supporting the determination of the correct number of lines of the image reproduced for the source image in the form of a picture from Figure 7d—menu of multifunctional device, frequency of the reveal emission signal $f_o = 235$ MHz, reception bandwidth $BW = 10$ MHz, correct number of lines $B_{Corr} = 288$.

**Table 4.** Normalized values (in relation to the maximum value) of the variability of measures as a function of the number of image lines, supporting the determination of the correct number of lines of the image reproduced for the source image in the form of a picture from Figure 7d.

| Number Lines | Method I | Method II | Method III | Method IV | Method V | Method VI | Method VII |
|---|---|---|---|---|---|---|---|
| 268 | 0.39560 | 0.72202 | 0.12345 | 0.13999 | 0.99729 | 0.17261 | 0.40321 |
| 269 | 0.38852 | 0.71628 | 0.11724 | 0.14128 | 0.99798 | 0.17480 | 0.40424 |
| 270 | 0.39586 | 0.72202 | 0.14957 | 0.16084 | 1.00000 | 0.20340 | 0.42164 |
| 271 | 0.39601 | 0.72202 | 0.10712 | 0.14628 | 0.99764 | 0.18310 | 0.40894 |
| 272 | 0.43179 | 0.76403 | 0.18302 | 0.17021 | 0.99415 | 0.22722 | 0.43696 |
| 273 | 0.41748 | 0.73856 | 0.13408 | 0.14586 | 0.99272 | 0.18691 | 0.41091 |
| 274 | 0.40313 | 0.72765 | 0.12056 | 0.14146 | 0.99411 | 0.17916 | 0.40655 |
| 275 | 0.41022 | 0.73316 | 0.13212 | 0.15153 | 0.99541 | 0.19758 | 0.41777 |
| 276 | 0.41721 | 0.73856 | 0.13409 | 0.15019 | 0.99722 | 0.19331 | 0.41593 |
| 277 | 0.39557 | 0.72202 | 0.12329 | 0.14326 | 0.99789 | 0.18120 | 0.40812 |
| 278 | 0.38845 | 0.71628 | 0.12115 | 0.13882 | 0.99803 | 0.17339 | 0.40374 |
| 279 | 0.36698 | 0.69831 | 0.12118 | 0.13987 | 0.99757 | 0.17336 | 0.40317 |
| 280 | 0.40309 | 0.72765 | 0.12922 | 0.14519 | 0.99934 | 0.18086 | 0.40614 |
| 281 | 0.42473 | 0.74386 | 0.11886 | 0.14716 | 0.99716 | 0.18588 | 0.40893 |
| 282 | 0.46779 | 0.78828 | 0.12500 | 0.15619 | 0.99354 | 0.19800 | 0.41511 |
| 283 | 0.44629 | 0.75913 | 0.12259 | 0.16599 | 0.99310 | 0.20466 | 0.41936 |
| 284 | 0.47511 | 0.79286 | 0.13992 | 0.18488 | 0.99487 | 0.22232 | 0.42804 |
| 285 | 0.46765 | 0.77355 | 0.12281 | 0.20281 | 0.99739 | 0.22925 | 0.43367 |
| 286 | 0.48921 | 0.80176 | 0.12755 | 0.24332 | 0.99752 | 0.26242 | 0.45277 |
| 287 | 0.55386 | 0.83804 | 0.10737 | 0.33133 | 0.99783 | 0.32414 | 0.49738 |
| 288 | 1.00000 | 1.00000 | 1.00000 | 1.00000 | 0.99721 | 1.00000 | 1.00000 |
| 289 | 0.54696 | 0.83428 | 0.10242 | 0.32685 | 0.99902 | 0.31612 | 0.49262 |
| 290 | 0.53273 | 0.82658 | 0.12042 | 0.24105 | 0.99907 | 0.25682 | 0.45063 |
| 291 | 0.46780 | 0.77355 | 0.13092 | 0.20498 | 0.99497 | 0.23382 | 0.43644 |
| 292 | 0.48941 | 0.78718 | 0.12532 | 0.18139 | 0.99280 | 0.21495 | 0.42547 |
| 293 | 0.46072 | 0.76883 | 0.11976 | 0.16428 | 0.99501 | 0.20192 | 0.41816 |
| 294 | 0.45344 | 0.77886 | 0.12064 | 0.15489 | 0.99647 | 0.19389 | 0.41349 |
| 295 | 0.46756 | 0.78828 | 0.12516 | 0.14966 | 0.99661 | 0.19051 | 0.41226 |
| 296 | 0.42435 | 0.75888 | 0.12496 | 0.14412 | 0.99943 | 0.18124 | 0.40757 |
| 297 | 0.37405 | 0.70442 | 0.12349 | 0.14030 | 0.99826 | 0.17469 | 0.40336 |
| 298 | 0.38852 | 0.71628 | 0.13211 | 0.14151 | 0.99718 | 0.18221 | 0.40808 |
| 299 | 0.43900 | 0.75414 | 0.12677 | 0.14409 | 0.99981 | 0.18282 | 0.40906 |
| 300 | 0.41032 | 0.73316 | 0.14947 | 0.15865 | 0.99753 | 0.21579 | 0.42951 |
| 301 | 0.40298 | 0.72765 | 0.11964 | 0.14368 | 0.99453 | 0.18400 | 0.40913 |
| 302 | 0.40303 | 0.72765 | 0.12215 | 0.14211 | 0.99284 | 0.18291 | 0.40781 |
| 303 | 0.41749 | 0.73856 | 0.12221 | 0.14278 | 0.99513 | 0.18106 | 0.40706 |
| 304 | 0.41013 | 0.73316 | 0.14081 | 0.15519 | 0.99662 | 0.20144 | 0.42054 |
| 305 | 0.43161 | 0.76403 | 0.10783 | 0.14623 | 0.99809 | 0.18605 | 0.41077 |
| 306 | 0.40996 | 0.73316 | 0.17973 | 0.17243 | 0.99893 | 0.22547 | 0.43573 |
| 307 | 0.40284 | 0.72765 | 0.13659 | 0.14814 | 0.99804 | 0.18706 | 0.41017 |
| 308 | 0.38140 | 0.71041 | 0.12275 | 0.14026 | 0.99934 | 0.17604 | 0.40453 |

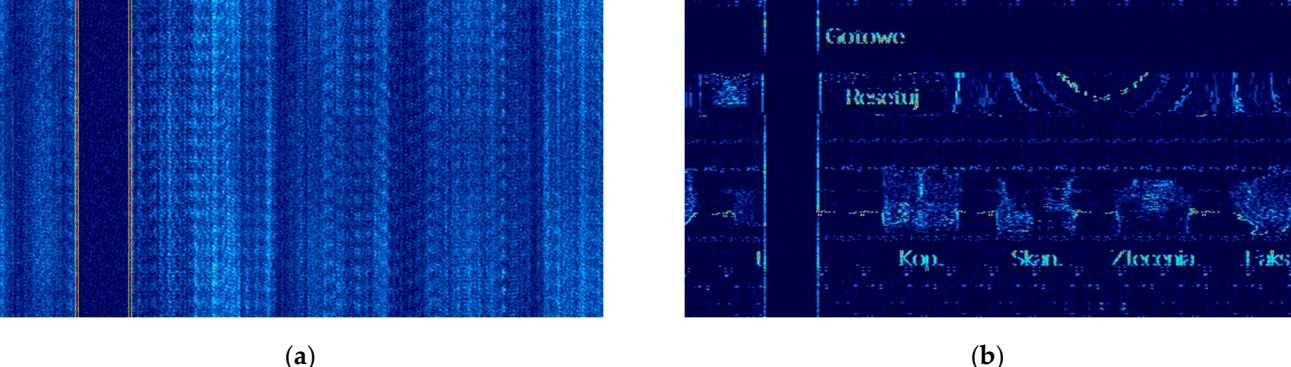

(**a**)             (**b**)

**Figure 18.** Images reconstructed from the revealing emission signal measured at the frequency $f_o = 235$ MHz, for the number of lines of this image determined in accordance with the criterion of the maximum value of the measures presented in Figure 17: (**a**) the number of image lines determined in accordance with method V ($B = 273$, the number of lines smaller than required), (**b**) number of image lines determined in accordance with method I, II, III, IV, VI, and VII ($B_{Corr} = 288$, correct number of lines).

### 4.4. The Analysis of Obtained Results

The analyses were carried out on the basis of test images presented in Figure 7, which were the sources of undesirable emissions during their processed in the graphic tracks of IT devices. Signals corresponding to these emissions were recorded and used in the rasterization process, i.e., their reconstruction also in the form of images.

The images presented in Figure 7 are only examples. The authors carried out several statistical tests for which other images were used (Figure 19). Obtained results for mentioned images only confirmed the conclusions stated below.

During conducted tests, different scenarios were adopted. DVI, HDMI, and VGA graphic standards, printer displays, and display of terminal VoIP were tested (Table 5). These allowed us to check proposed methods and conventional methods from the viewpoint of suitability for determination of the number lines of reconstructed images. Results of detailed analyses are presented for images from Figure 7.

For the process to be successful, however, the basic raster parameters are necessary, which are the length $d$ of the image lines (image width) and the $B$ number of image lines (image height). At the beginning, an assumption was made about the knowledge of the image line length $d$ and the pre-estimated number $B_{Entr}$ of the image line, which was carried out on the basis of the analysis of amplitude time variability of the revealing emission signal. The authors' experience shows that the accuracy of the rough calculation of the number of lines in the image is $\pm 20$ lines. The rough estimation of the $B_{Entr}$ parameter allowed for the estimation of $B_{Corr}$ based on the methods of contrast evaluation (methods I, II, III, and IV) and the methods proposed by the authors of the article (methods V, VI, and VII). Taking into account the accuracy of the rough estimation of $B_{Entr}$ quantity, the maximum value of the measure, calculated in the variability range $(B_{Entr} \pm n)$, where $n = 0, 1, 2, \ldots, 20$, was adopted as the criterion for the correctness of determining the number of image lines. This means that the image height was decreased and increased in increments of 1, up to a maximum of 20 lines. As a result of the performed calculations of the values of the adopted measures (methods I to VII), the appropriate number of $B_{Corr}$ lines for each reproduced image was determined (Table 6).

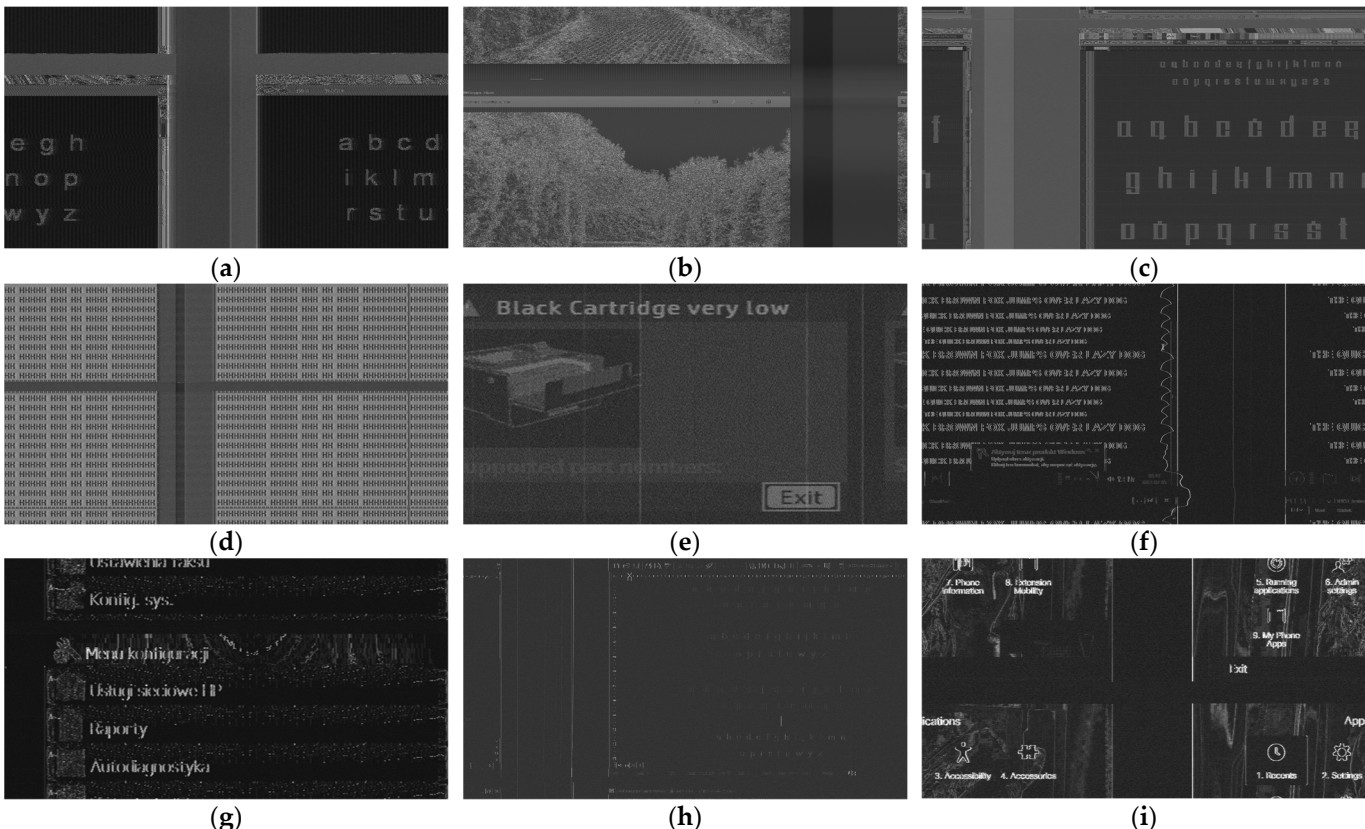

**Figure 19.** Images for which statistical analyses were conducted (images reconstructed on base on reveal emissions for thirty times summation without colorization): (**a**) DVI standard, receive frequency $f_o = 365$ MHz, $BW = 50$ MHz, number lines $B_{corr} = 525$, resolution $640 \times 480/60$ Hz; (**b**) DVI standard, receive frequency $f_o = 1334$ MHz, $BW = 50$ MHz, number lines $B_{corr} = 1066$, resolution $1280 \times 1024/60$ Hz; (**c**) DVI standard, receive frequency $f_o = 1805$ MHz, $BW = 100$ MHz, number lines $B_{corr} = 1066$, resolution $1280 \times 1024/60$ Hz; (**d**) DVI standard, receive frequency $f_o = 1775$ MHz, $BW = 100$ MHz, number lines $B_{corr} = 1125$, resolution $1920 \times 1080/60$ Hz; (**e**) laser printer HP M507, menu with icons, receive frequency $f_o = 392$ MHz, $BW = 10$ MHz, number lines $B_{corr} = 266$; (**f**) VGA standard, receive frequency $f_o = 450$ MHz, $BW = 20$ MHz, number lines $B_{corr} = 628$, resolution $800 \times 600/60$ Hz; (**g**) laser printer, menu with text, receive frequency $f_o = 235$ MHz, $BW = 10$ MHz, number lines $B_{corr} = 288$; (**h**) DVI standard, receive frequency $f_o = 740$ MHz, $BW = 50$ MHz, number lines $B_{corr} = 628$, resolution $1280 \times 1024/60$ Hz; and (**i**) display of terminal VoIP, receive frequency $f_o = 800$ MHz, $BW = 20$ MHz, number lines $B_{corr} = 528$.

Considering the effectiveness of the adopted methods, however, a discussion should be held on the dynamics of changes in the calculated values of the adopted measures in accordance with the relationships (2), (8), (11), (14), (22), (26), and (29). Value of differences between the maximum and minimum value (example notation for method I):

$$Difference_I = maximum_I - minimum_I \qquad (38)$$

where

$$minimum_I = \min_B(Contrast_I(B)), \qquad (39)$$

may testify that the method is resistant to possible disturbances in the reproduced images (Table 2). The second important parameter of the assessment is the variance $\sigma^2$ (example notation for method I):

$$\sigma^2 = \frac{1}{41} \sum_B \left( Contrast_{I\_Un}(B) - \overline{Contrast_{I\_Un}} \right)^2, \tag{40}$$

where

$$\overline{Contrast_{I\_Un}} = \frac{1}{41} \sum_B Contrast_{I\_Un}(B). \tag{41}$$

**Table 5.** Parameters of sources of reveal emissions used in the tests.

| Source of Reveal Emission | Duration of Displayed Image | Frequency of Reveal Signal Emission | Bandwidth | Number Lines |
|---|---|---|---|---|
| Display of VoIP terminal—menu in form of icons | Unknown | 800 MHz | 20 MHz | 528 |
| Display of HP laser printer M477fdn—menu in form of text | Unknown | 235 MHz | 10 MHz | 288 |
| Display of HP laser printer M477fdn—menu in form of text | Unknown | 235 MHz | 10 MHz | 288 |
| Display of HP laser printer M507—menu in form of icons | Unknown | 392 MHz | 10 MHz | 266 |
| HDMI standard | $1280 \times 1024/60$ Hz | 1334 MHz | 50 MHz | 1066 |
| | $1280 \times 1024/60$ Hz | 200 MHz | 100 MHz | 1125 |
| | $1280 \times 1024/60$ Hz | 1334 MHz | 50 MHz | 1066 |
| | $1280 \times 1024/60$ Hz | 740 MHz | 50 MHz | 628 |
| DVI standard | $1920 \times 1080/60$ Hz | 1775 MHz | 100 MHz | 1125 |
| | $1280 \times 1024/60$ Hz | 1805 MHz | 100 MHz | 1066 |
| | $640 \times 480/60$ Hz | 365 MHz | 50 MHz | 525 |
| VGA standard | $800 \times 600/60$ Hz | 450 MHz | 20 MHz | 628 |
| | $1024 \times 768/60$ Hz | 558 MHz | 10 MHz | 806 |

**Table 6.** $B_{Entr}$ input data and the determined $B_{Corr}$ values for the adopted criterion of maximizing the value of the calculated measure for the example images shown in Figure 7.

| | Figure 7a | | Figure 7b | | Figure 7c | | Figure 7d | |
|---|---|---|---|---|---|---|---|---|
| | $B_{Entr}$ | $B_{Corr}$ | $B_{Entr}$ | $B_{Corr}$ | $B_{Entr}$ | $B_{Corr}$ | $B_{Entr}$ | $B_{Corr}$ |
| Method I | | 1066 | | 1125 | | 806 | | 288 |
| Method II | | 1066 | | 1125 | | 823 | | 288 |
| Method III | | 1066 | | 1125 | | 806 | | 288 |
| Method IV | 1069 | 1066 | 1125 | 1125 | 809 | 806 | 294 | 288 |
| Method V * | | 1062 | | 1145 | | 793 | | 270 |
| Method VI * | | 1066 | | 1125 | | 806 | | 288 |
| Method VII * | | 1066 | | 1125 | | 806 | | 288 |

*—method used to estimate the length d line of the reconstructed image [28].

The preliminary analysis of the obtained results shows that method II, and in particular method V, are not measures allowing for the correct determination of the $B_{Corr}$ number of the image line (Tables 7 and 8). For each of the analysed images, method V indicated the wrong number of image lines. Method II turned out to be ineffective only in the case of the image presented in Figure 7c, which allows for its rejection anyway. The other methods, i.e., method I, III, IV, VI, and VII and the accepted criteria of acceptability, correctly indicated the $B_{Corr}$ number of the lines of the reproduced images. However, due to the values presented in Tables 2 and 3 and the analysis of the sensitivity of the methods to the poor quality of the reproduced images ($\sigma^2 < 0.01$, $Difference < 0.5$), methods III and VI can be indicated as

effective in determining the number of lines of the image obtained in the electromagnetic infiltration process.

**Table 7.** Differences between the maximum and minimum values calculated according to (2), (8), (11), (14), (22), (26), and (28).

|  | **Figure 7a** | **Figure 7b** | **Figure 7c** | **Figure 7d** |
|---|---|---|---|---|
| Method I | 0.51890 | 0.44351 | 0.06312 | 0.63302 |
| Method II | 0.33632 | 0.11059 | 0.04089 | 0.30169 |
| Method III | 0.88059 | 0.86782 | 0.74212 | 0.89758 |
| Method IV | 0.62317 | 0.62085 | 0.53086 | 0.86118 |
| Method V | 0.00488 | 0.00399 | 0.00577 | 0.00728 |
| Method VI | 0.68045 | 0.75682 | 0.76299 | 0.82739 |
| Method VII | 0.28928 | 0.45506 | 0.61874 | 0.59683 |

**Table 8.** Values of the variance of the differences between the maximum and minimum values calculated according to (2), (8), (11), (14), (22), (26), and (28).

|  | **Figure 7a** | **Figure 7b** | **Figure 7c** | **Figure 7d** |
|---|---|---|---|---|
| Method I | 0.0114988 | 0.0063718 | 0.0002526 | 0.0096783 |
| Method II | 0.0067733 | 0.0003845 | 0.0000599 | 0.0026803 |
| Method III | 0.0182098 | 0.0174844 | 0.0128797 | 0.0183419 |
| Method IV | 0.0129926 | 0.0104123 | 0.0109060 | 0.0184546 |
| Method V | 0.0000013 | 0.0000012 | 0.0000029 | 0.0000043 |
| Method VI | 0.0221255 | 0.0183717 | 0.0158086 | 0.0162905 |
| Method VII | 0.0039669 | 0.0050373 | 0.0101824 | 0.0084351 |

The choice of an effective method in the determining the number $B_{Corr}$ of lines in the image reconstructed in the electromagnetic infiltration process was based on calculating the differences between the maximum and minimum values of individual measures calculated for images with the number of lines equal to $B$ (Table 7). In order to improve the correctness of the selection, the assessment of the analysed methods was also based on the variance $\sigma^2$ values of individual measures calculated as a function of parameter $B$ (Table 8). The obtained results allowed to indicate methods III and VI as effective methods in the determining the correct $B_{Corr}$ number. When analyzing the values of the variance $\sigma^2$ and the mentioned differences of the maximum and minimum values, one may wonder whether the methods I, IV, and VII cannot also be used to determine the number of lines in the image. The values of the measures under consideration clearly indicate, through the maximum value, the correct number $B_{Corr}$. However, the distance between the minimum values and the maximum may, according to the authors, be insufficient in practice for other images reconstructed from the emission revealing signals.

## 5. Conclusions

This article presented the issue related to the correct determination of the number $B_{Corr}$ of the reproduced image lines on the basis of the recorded revealing emission signal. Determining the correct $B_{Corr}$ value is very important when it is necessary to further process the image using the coherent summation method in order to improve its quality, i.e., improve the signal-to-noise (*SNR*) parameter. An incorrectly determined number $B_{Corr}$ of lines of the reproduced image causes the summing up of several dozen repetitions of the same image, reproduced from a sufficiently long implementation of the revealing emission signal, resulting in blurring and not sharpening the data contained in the image.

Four methods of contrast assessment and three methods proposed by the authors of the article were used in the analyses. The latter were successfully used in determining the line length of the reconstructed image.

In the conducted analyses, recorded, real emission signal revealing and reconstructed images on their basis were used (Figure 7). The sources of these emissions were the

graphic lines (HDMI/DVI and VGA standards) of the computer system, the display of the multifunctional devices and the display a of VoIP terminal. Images with different graphic structures were displayed on a computer monitor, which allowed for the assessment of the considered methods in terms of their effectiveness for various scenarios.

Analysing the considered methods from I to VII, one can notice the usefulness of methods III and VI in the process of determining the number of lines for the reproduced image. Method III is a typical method of determining image contrast; method VI is the method proposed by the authors of the article (Table 9). Is it therefore necessary to present new methods, since the known method used in determining the contrast of the image is effective in the process of determining the correct number of lines in the image? There is only one answer to this question. Yes. It is enough to analyse the complexity of the calculations necessary to be carried out by all abovementioned methods and thus the time required to perform the necessary calculations. Method III is based on calculating the squares of the differences of the relevant quantities, which then must be summed up many times. Method VI requires the calculation of only the sums of the given maximum values. Hence, undoubtedly, the proposed method VI has an advantage over the conventional method, and it is proposed to be used in the process of electromagnetic infiltration, in which time plays a very important role.

**Table 9.** Evaluation of methods.

| Method | Useless | Useless Due to Sensitivity to Disturbances | Useful | Useful Due to Low Computational Complexity |
|---|---|---|---|---|
| Conventional methods | | | | |
| Method I | | X | | |
| Method II | X | | | |
| Method III | | | X | |
| Method IV | | X | | |
| Method proposed by authors | | | | |
| Method V | X | | | |
| Method VI | | | X | X |
| Method VII | | X | | |

Further work in this area will focus on the software implementation of the algorithms of the proposed methods, allowing for the automation of the process of determining the $B_{Corr}$ number of lines of the reproduced image. This process is to be associated with algorithms for estimating the line length d of the image, which should ultimately accelerate the activities related to the correct reproduction of graphic data and making correct decisions on the classification of electromagnetic emissions.

**Author Contributions:** Conceptualization, I.K. and A.P.; methodology, I.K., A.P. and K.G.; software, I.K.; validation, I.K.; formal analysis, I.K., A.P. and K.G.; investigation, I.K.; resources, I.K. and A.P.; writing—original draft preparation, I.K.; writing—review and editing, I.K. and A.P.; visualization, I.K.; supervision, A.P. and K.G.; project administration, I.K. All authors have read and agreed to the published version of the manuscript.

**Funding:** This research received no external funding.

**Institutional Review Board Statement:** Not applicable.

**Informed Consent Statement:** Not applicable.

**Data Availability Statement:** Not applicable.

**Conflicts of Interest:** The authors declare no conflict of interest.

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
