# Peer review of "Number of Lines of Image Reconstructed from a Revealing Emission Signal as an Important Parameter of Rasterization and Coherent Summation Processes"

_applsci, doi:10.3390/app13010447_

Round 1

Reviewer 1 Report

     In this manuscript, the so called Number of lines of image reconstructed from a revealing emission signal as an important parameter of rasterization and coherent summation processes is analyzed and discussed.

Nevertheless, the commented key issues must be re emphasized to the authors, due to the low content quality of this eidtion.

  (1)  In the abstract,  no any  numerical gain for the proposed scheme to determine the correct number of lines is included.  The authors must provide the several key performance improvement to identify  the superiority of the developing scheme compared to the well discussed conventonal solutions.

  (2) The conclusion is too long to illustrate the main contribution of this work. The several distinct numerical results must be presented  in the conclusion, compared to the abstract.

 (3)  The table of all parameters concerned in the test must be included in the modification of this work and the reference and  setting support of all parameters and the datasheets must be labelled to prove the physical implementability.

  (4) For clarity, the flow chart of the propose schemes must be included and compared with the conventional ones.

  (5) The figures for all the concened test scenarios and related devices must be included in this manuscript.

  (6) All concenred devices provide the VGA, HDMI interfaces must be presented in figures as well.

  (7) For vertify the reliability of the numerical results, the repetitive experiments must be made and attached to this work to prove the repeatiblity of all the performance characteristics    with statistical analysis.

Before all above modifications are made, I cannot recommend this manuscript to publish. 

Author Response

Dear Reviewer

Thank you very much for your reviewing. We incredibly appreciate all your positive and constructive comments and suggestions on our manuscript. All our comments are below. Changes or additions to the manuscript are indicated by red and blue colors font.

  1. In the abstract,  no any  numerical gain for the proposed scheme to determine the correct number of lines is included.  The authors must provide the several key performance improvement to identify  the superiority of the developing scheme compared to the well discussed conventional solutions.

Thank you for the suggestion. The paper was completed and we added four tables (1, 2, 3 and 4). The tables include number values which apply to Figures 11, 13, 15 and 17. We also written several sentences in Conclusions which explain why we propose new approach to solution the problem of determination of number lines.

  1. The conclusion is too long to illustrate the main contribution of this work. The several distinct numerical results must be presented  in the conclusion, compared to the abstract.

Thank you. You are right. The Conclusions is very long and the part of the paper includes a lot information. Now we rebuilt this part and we put very short information about usefulness of tested methods (in form of table 8)

  1. The table of all parameters concerned in the test must be included in the modification of this work and the reference and  setting support of all parameters and the datasheets must be labelled to prove the physical implement ability.

Thank you for this remark. We forgotten about this very important thing which could allow to repeat such tests by other researchers. New Table 5 in chapter 4.4 includes information about parameters of tested devices. Also examples of measuring systems were put in the chapter 2.2.

  1. For clarity, the flow chart of the propose schemes must be included and compared with the conventional ones.

To understand how values of considered methods are calculated, we built an algorithm of the determining the correct number lines of the reconstructed image. The algorithm explains step by step the procedure which aim is determining correct number lines of reconstructed images (Figure 9, chapter 2.3).

  1. The figures for all the concerned test scenarios and related devices must be included in this manuscript.
  2. All concerned devices provide the VGA, HDMI interfaces must be presented in figures as well.

Of course the remark is very important. Such pictures can show a location each devices during conducted tests. Some pictures the chapter 2.2 includes. The pictures show interfaces which were under tests and pictures of laser printers.

  1. For vertify the reliability of the numerical results, the repetitive experiments must be made and attached to this work to prove the repeatiblity of all the performance characteristics    with statistical analysis.

Thank you for the significant remark. We were tested a lot of sources of reveal emissions. The sources are connected with digital and analog interfaces (HDMI, DVI, VGA), VoIP terminal, laser printers. The tests allowed to confirm the results presented in this paper. The chapter 4.4 includes information about statistical tests.

Finally, we thank you again for careful review and valuable suggestions, which is very helpful for improving the quality of the manuscript.

Yours sincerely,

All the authors.

Reviewer 2 Report

The paper is a very interesting one, in a very difficult area. The presentation is good and also the references are new or very new.

It is room of improvement regarding the possible conclusions is we correlate the results of the analyzed methods.

An important part of the paper is related to the different methods to reconstruct the images from the captured signals. Section 3.2 is dedicated to the methods for images reconstruction based on the contrast evaluation and section 3.3 is related to similar methods proposed by papers authors. In total there are 7 methods. For all of them a reference is cited to help in founding more information. And also all of them are presented using a similar pattern, so for this point of view the paper looks very well organized.

Despite to this, in order to increase the paper readability it is very useful to add for each method very briefly some relevant details and explanations: what is the expected performance and the main achievement for the presented method, what is the element who underline each method and how relevant (or not) it is, etc. Maybe this may be highlighted also in the area of the test results (chapter 4), but it is good to introduce also a kind of diversity in the 6-pages-length area describing the methods (chapter 3) to increase the overall paper readability. 

Author Response

Dear Reviewer

Thank you very much for your reviewing. We incredibly appreciate all your positive and constructive comments and suggestions on our manuscript. We tried to correct our paper according to your remarks. We hope that now the paper sounds better and potential readers will appreciate our work.

Changes or additions to the manuscript are indicated by red and blue colors font.

Yours sincerely,

All the authors.
